# Algorithm Development in Neural Networks:
# Insights from the Streaming Parity Task

**Loek van Rossem** [1]   **Andrew M. Saxe** [1,2]

## Abstract

Even when massively overparameterized, deep neural networks show a remarkable ability to generalize. Research on this phenomenon has focused on generalization within distribution, via smooth interpolation. Yet in some settings neural networks also learn to extrapolate to data far beyond the bounds of the original training set, sometimes even allowing for infinite generalization, implying that an algorithm capable of solving the task has been learned. Here we undertake a case study of the learning dynamics of recurrent neural networks (RNNs) trained on the streaming parity task in order to develop an effective theory of algorithm development. The streaming parity task is a simple but nonlinear task defined on sequences up to arbitrary length. We show that, with sufficient finite training experience, RNNs exhibit a phase transition to perfect infinite generalization. Using an effective theory for the representational dynamics, we find an implicit representational merger effect which can be interpreted as the construction of a finite automaton that reproduces the task. Overall, our results disclose one mechanism by which neural networks can generalize infinitely from finite training experience.

## 1. Introduction

Examples of computational algorithms appearing in deep networks are numerous (Olah et al., 2020; Goh et al., 2021; Wang et al., 2022; Power et al., 2022; Nanda et al., 2023; Zhong et al., 2023). Furthermore, recurrent neural networks and transformers have shown a noteworthy ability to generalize (Loula et al., 2018; Lake & Baroni, 2018; Brown et al.,

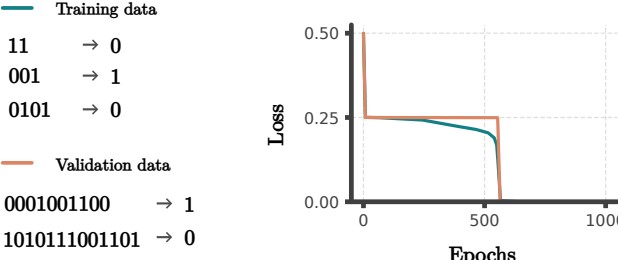

Figure 1. Sequence length generalization of a recurrent network on a simple task. **Left**: Example sequences of the streaming parity task, from the training dataset (short sequences) and the validation dataset (long sequences). **Right**: The mean squared loss of a recurrent neural network during training on the streaming parity task. Note that the loss also goes to zero for sequences much longer than found in the training data. This was tested for sequences up to length 10000.

2020) in particular to sequences with lengths not seen in the training data (Dai et al., 2022; Abbe et al., 2023; Cohen-Karlik et al., 2023). These results are surprising as gradient descent provides no clear incentive to generalize beyond the training domain. Why do deep learning systems sometimes develop proper computational algorithms, instead of simply interpolating the data? Understanding this is crucial for the safe application of machine learning models, as for instance, models can appear to generalize initially, but break after moving too far away from the training domain (Anil et al., 2022; Zhou et al., 2024). Similarly, in neuroscience, an important goal is to try to figure out the types of algorithms the brain employs. This is particularly difficult since the brain is so complex, interconnected, and poorly understood (Thompson & Best, 1989; Olshausen & Field, 2006) that it is unclear how to even recognize an algorithm when we see it. Instead, studying the dynamics of algorithm development in the brain might be more tractable (Richards et al., 2019), as learning rules may not be as complex as the learned algorithm itself. Despite attempts to understand these dynamics (Zhou et al., 2021; El-Gaby et al., 2024), it is still a highly challenging problem by itself. Here we hope to provide a better sense of what to look for in the brain, by answering analogous questions in a simpler setting.

[1]Gatsby Computational Neuroscience Unit, University College London [2]Sainsbury Wellcome Centre, University College London. Correspondence to: Loek van Rossem <loek.rossem.22@ucl.ac.uk>.

*Proceedings of the 42ⁿᵈ International Conference on Machine Learning*, Vancouver, Canada. PMLR 267, 2025. Copyright 2025 by the author(s).

Let us consider a relatively simple computational problem for which algorithm development can still be studied: the streaming parity task (Figure 1). Given a sequence of zeros and ones with varying length, the aim of the task is to output a zero when the number of ones is even and output a one when the number of ones is odd. A recurrent neural network trained only on sequences up to some short finite length will sometimes generalize infinitely. It is able to solve the task accurately for any sequence length no matter how large, even for sequences thousands of times longer than shown in the training data.

As longer sequences are not within the same domain as shorter sequences, the generalization cannot simply be explained by interpolation. One can continue feeding in symbols and the network will continue predicting correctly, suggesting it has somehow learned a computational algorithm. It is unclear how such an algorithm can develop during training from gradient descent optimization. The network is trained to reduce its loss only on the shorter sequence dataset; it is not penalized for breaking after a certain length.

The main goal of this paper is to find simple mathematical models able to explain this seemingly surprising behavior. Our main contributions are as follows:

- In Section 3, we provide a local interaction theory for representational learning dynamics in recurrent neural networks.

- In Section 4, we explain how these interactions can result in the development of an algorithm capable of out-of-distribution generalization.

- In Section 5, we find algorithm development occurs in two phases: an initial tree fitting phase, and a secondary generalization phase.

## 2. Automata and Recurrent Neural Networks

### 2.1. Interpreting Recurrent Neural Networks

The first step to understanding the development of computational algorithms in recurrent neural networks, is to choose the right way of representing the information that defines the network. The model's parameters are a complete but poor representation. They also contain a large amount of redundant information, e.g. swapping the order of neurons does not affect the encoded algorithm in any way.

For analyzing the inner structure of neural networks a better approach is to consider the geometry of the representational structure, i.e. how are the hidden activations corresponding to the data structured in the network (Lin et al., 2019; Williams et al., 2022; Lin & Kriegeskorte, 2023). This is however, still not ideal in the context of computational algorithms. Although understanding how data is represented in

the network may be helpful, it does not directly say much about the nature of the computations being performed on that representational space. Moreover, the representational geometry can vary greatly across RNN architectures trained on the same task (Maheswaranathan et al., 2019).

One common approach to interpret RNNs is with dynamical systems (Sussillo & Barak, 2013; Laurent & Brecht, 2016; Can et al., 2020; Driscoll et al., 2024). Here, due to our focus on computational algorithms, we will instead be interpreting the recurrent neural network using deterministic finite automata (DFA), an approach also well known in the RNN literature (Servan-Schreiber et al., 1989; Giles et al., 1992; Omlin & Giles, 1996; Tino et al., 1999; Weiss et al., 2020; Merrill & Tsilivis, 2022; Michaud et al., 2024), which has also seen some applications in a neuroscience context (Turner et al., 2021; Brennan et al., 2023).

A DFA consists of a set of states including an initial state, transitions between the states given input symbols, and an output symbol for each state. An example of such a DFA solving the streaming parity task can be seen in Figure 2.

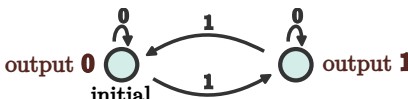

*Figure 2.* Example of a two state DFA solving the streaming parity task. Every time it receives a 1 as an input it alternates between the state that will output 0 and the state that will output 1. When it receives a 0 it remains in its current state.

### 2.2. Automaton Extraction

We will use a relatively simple method for constructing an automaton from the representational space of an RNN, as this will be enough for our purposes. Consider an abstract recurrent neural network:

$$
\begin{aligned}
h_t &= f_h(h_{t-1}, x_t) \\
y_t &= f_y(h_t)
\end{aligned}, \tag{1}
$$

where the exact forms of the recurrent map $f_h$ and output map $f_y$ will depend on the architectural details of the network.

From the hidden representations of this network we can extract a deterministic finite automaton. States are defined to be the hidden representations after receiving each sequence, transitions are determined by following where states go after the network received an input symbol, and outputs are simply the network's output map evaluated at each state. This procedure is illustrated in Figure 3.

When two different sequences of input symbols are assigned the same internal activation vector, an additional application

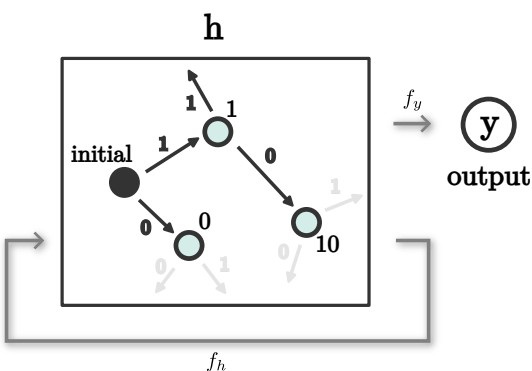

*Figure 3.* Example of the automaton extraction procedure. Before the network has seen any input symbols the internal vector of hidden activations has some initial value $h_0$, which we will call the initial state of the automaton. After seeing a symbol, say e.g. 1, the hidden activations change to a different vector $f_h(h_0, 1)$. We will call this vector the state corresponding to input sequence 1, and say that the initial state has a transition to the state 1 after receiving the input symbol 1. We can repeat this procedure for all possible sequences, to define a state for each one and transitions on each state for every possible input symbol. We can assign an output to each state by evaluating the output map on its corresponding representation.

of the recurrent map will send them to the same internal activation vector for any possible input symbol received. Their activations will remain the same in the future, and the output map will always assign them the same output from that point on. They can no longer be distinguished, so we will consider them to be the same state in the automaton.

If enough representations overlap, it may be the case that all transitions go to already existing states. At this point, we have a finite set of states capable of representing all possible internal states of the network. These states, together with the transitions between them and their outputs, form a discrete computational algorithm capable of producing outputs on input sequences, which match the outputs of the RNN. For more details on automaton extraction, see Appendix A.

### 2.3. Automata During Training

In order to visualize the development of an algorithm, we extract automata at each epoch from an RNN during training on the streaming parity task (Figure 4). We can see that, due to random small weights, the automaton initially has few states and random outputs. As the model trains, the automaton expands into a complete tree fitting the training data. Then, right when the training loss becomes zero, states in the automaton appear to merge until it becomes finite, and we see generalization on all data. To understand this better, we will first try to model the representational dynamics, and then study the induced dynamics on the automaton.

## 3. Implicit State Merger

### 3.1. Intuition

Why would states merge during training? The representational space is typically high-dimensional, so it seems statistically unlikely that many different representations end up at the same vector by chance. The key insight here is that, due to continuity, sometimes the fastest way for gradient descent to minimize the loss is to merge nearby representations.

As an example for illustration, suppose that at some point during training, two sequences in the dataset agree on target outputs, and one already has the correct predicted output. Then, if the recurrent map $f_h$ adjusts to move the other representation closer, its target output will also move towards the correct prediction, as the output map $f_y$ is continuous. Continuity thus gives rise to an interaction effect between nearby representations, potentially resulting in merging.

Implicit bias from gradient descent is a well-studied topic in the deep learning literature (Neyshabur et al., 2015; Gunasekar et al., 2018; Chizat & Bach, 2020; Soudry et al., 2022). However, the relatively simple effect we are considering here will turn out to be particularly interesting in the context of algorithm development in RNNs.

### 3.2. Interaction Model

Let us attempt to formalize this intuition by modeling the interaction between two nearby states, using the modeling approach from (van Rossem & Saxe, 2024), adapted for recurrent networks. Suppose that we have two input sequences $x_1^{(1)} \cdots x_1^{(m_1)}$ and $x_2^{(1)} \cdots x_2^{(m_2)}$ in our dataset, with corresponding hidden representations $h(x_1^{(1)} \cdots x_1^{(m_1)})$, $h(x_2^{(1)} \cdots x_2^{(m_2)})$. We would like to understand the behavior of these representations when they get near each other during training, in an arbitrary neural network.

**Arbitrary architecture with high expressivity.** Instead of analyzing this interaction for a specific recurrent architecture, we model it for an arbitrary network with high *expressivity*, meaning any network large and complex enough to have the freedom to behave like a smooth map. Under this assumption, we replace the effect of the assignment of hidden states by the network's parameters with two arbitrarily optimizable vectors $h_1, h_2$, and replace the effect of the network assigning output prediction to those hidden states with arbitrarily optimizable smooth maps $y_1, \ldots, y_N$ (schematized in Figure 5). Note that because we are dealing with a recurrent architecture, there may be multiple data points which have $h(x_1^{(1)} \cdots x_1^{(m_1)})$ or $h(x_2^{(1)} \cdots x_2^{(m_2)})$ as intermediate hidden states. Thus, we have $N$ output maps, one for each possible sequence after $x_1^{(1)} \cdots x_1^{(m_1)}$ and $x_2^{(1)} \cdots x_2^{(m_2)}$.

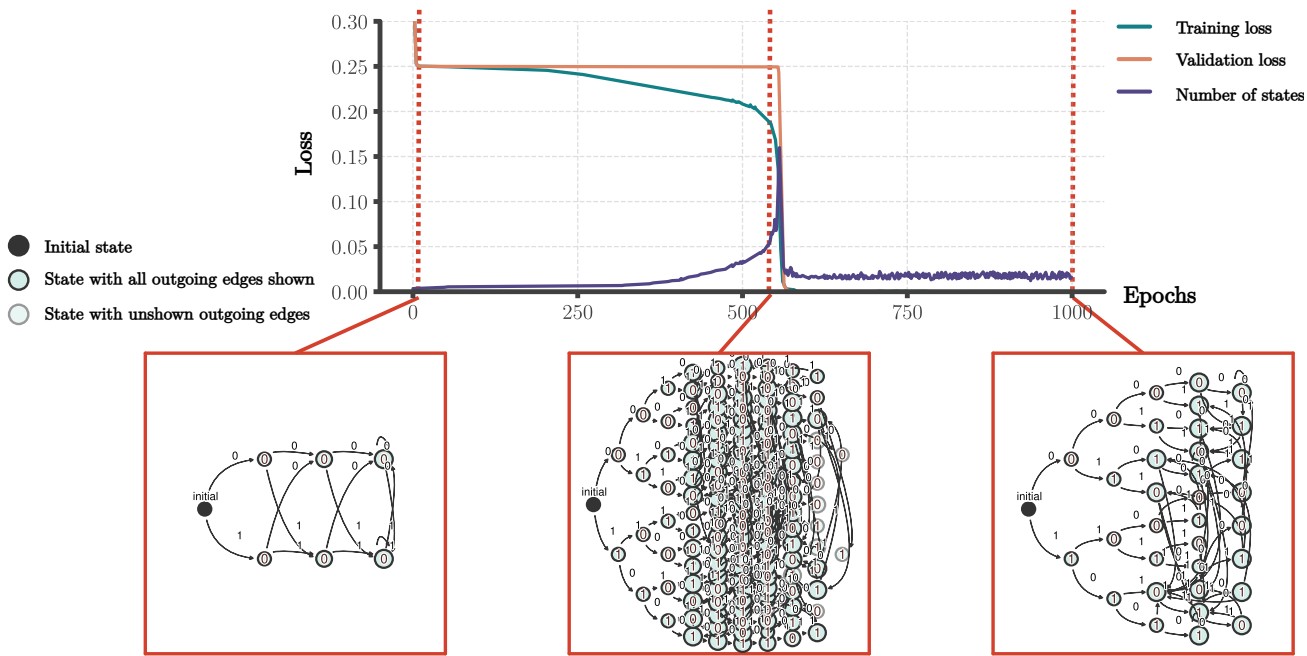

*Figure 4.* Automata extracted from the hidden layer of an RNN during training on the streaming parity task. First the model fits the training data with a complete tree. After that states merge until the automaton becomes finite, which is when generalization occurs. The RNN consisted of a single fully connected recurrent layer with 100 units and ReLU activation. It was trained on all examples of the streaming parity task, up to length 10.

To simplify the theoretical analysis, we assume here that any sequence $x_1^{(1)} \cdots x_1^{(m_1)} \tilde{x}^{(1)} \cdots \tilde{x}^{(n_i)}$ in the dataset has a matching sequence $x_2^{(1)} \cdots x_2^{(m_2)} \tilde{x}^{(1)} \cdots \tilde{x}^{(n_i)}$ and vice versa, as we do not expect pairs for which this is not the case to contribute to the interaction.

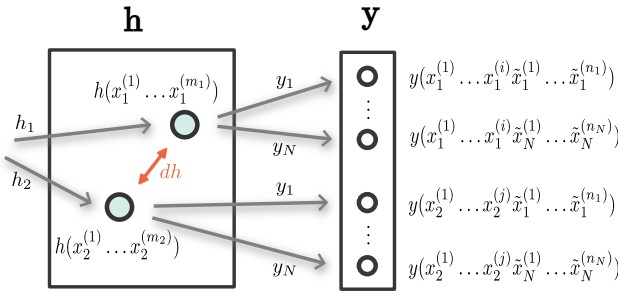

*Figure 5.* Schematic illustration of the interaction model between two nearby representations corresponding to sequences $x_1^{(1)} \cdots x_1^{(m_1)}$ and $x_2^{(1)} \cdots x_2^{(m_2)}$. The effect of the model assigning their representations is abstracted away with arbitrarily optimizable vectors $h_1$ and $h_2$, and the effect of the model assigning outputs for every of the $N$ subsequent sequences in the training data is abstracted away with arbitrarily optimizable smooth maps $y_1, \ldots, y_N$.

We will take gradients with respect to these arbitrary smooth maps. While a neural network may not optimize with smooth map dynamics and the gradient will depend on the architectural details, universal approximation theorems (Hornik et al., 1989; Csáji, 2001) tell us that an expressive one is at least able to freely model smooth maps. Therefore, we are modeling a simple behavior that, at least in principle, any expressive network can exhibit. The exact details of the architecture are not required to understand the key mechanisms by which the network learns to solve the task. Abstracting them away may help to find simpler and more intuitive explanations.

Note also that the maps $h_1, h_2, y_1, \ldots y_N$ may share parameters. They are not independent smooth maps. However, if the network is expressive enough, it can still have enough freedom to effectively optimize each map independently at specific points, so we will choose to ignore potential interaction effects arising from parameter sharing.

**Local linear approximation.** As we are trying to model the interaction between two representations, we will consider the case where the distance $dh := h_2 - h_1$ between them is small. We can thus take linear approximations of the maps $y_1, \ldots, y_N$ around the representational mean $\frac{1}{2}(h_2 + h_1)$:

$$y(x_\alpha^{(1)} \cdots x_\alpha^{(n_\alpha)} \bar{x}_i^{(1)} \cdots \bar{x}_i^{(m_i)}) \approx \bar{y}_i + \frac{1}{2} D_{y_i}(h_\alpha - h_{\neg\alpha}),$$
$$(2)$$

where $\alpha \in 1, 2$ and $i \in 1, \cdots, N$.

In general, the dynamics are undefined without specifying the parameterization. However, for the local linearized system, there is a unique choice of parameter-independent dynamics, namely by optimizing with respect to the effective linear parameters of the network $h_1, h_2, \bar{y}_1, \ldots, \bar{y}_N, D_{y_1}, \ldots, D_{y_N}$. For the mean squared loss

$$L = \frac{1}{2} \langle ||\bar{y}_i + \frac{1}{2} D_{y_i}(h_\alpha - h_{\neg\alpha}) - y_{\alpha,i}^*||^2 \rangle_{\alpha=1,2,i=1\ldots,N},$$
$$(3)$$

after taking the continuous-time limit and considering solutions where representations either move closer or further away form each other, we find the self-contained 3-scalar system

$$\frac{\mathrm{d}}{\mathrm{d}t}||dh||^2 = -\frac{1}{2}\frac{1}{\tau_h}\langle w_i \rangle_i$$

$$\frac{\mathrm{d}}{\mathrm{d}t}\langle||dy_i||^2\rangle_i = -\frac{1}{2}(\frac{1}{N\tau_y}||dh||^2 + \frac{1}{\tau_h}\frac{\langle||dy_i||^2\rangle_i}{||dh||^2})\langle w_i\rangle_i$$

$$\frac{\mathrm{d}}{\mathrm{d}t}\langle w_i\rangle_i = -\frac{1}{4}\frac{1}{N\tau_y}(3\langle w_i\rangle_i - \langle||dy_i||^2\rangle_i$$
$$+ \langle||y_{2,i}^* - y_{1,i}^*||^2\rangle_i)||dh||^2$$
$$- \frac{1}{4}\frac{1}{\tau_h}\frac{\langle w_i\rangle_i}{||dh||^2}(\langle||dy_i||^2\rangle_i + \langle w_i\rangle_i),$$
$$(4)$$

where $||dh||^2$ is the squared representational distance, $\langle||dy_i||^2\rangle$ is the average squared prediction distance, and $\langle w_i\rangle_i := \langle||dy_i||^2 - dy_i^\top(y_{2,i}^* - y_{1,i}^*)\rangle_i$ the average of an output alignment metric. The constants $1/\tau_h$ and $1/\tau_y$ are the *effective* learning rates of the representational map and output map respectively, as we are now optimizing with respect to these smooth maps as opposed to the model's parameters. The derivation and more details can be found in Appendix B.1.

Note that because we are considering the continuous-time limit, no noise has been introduced, and we also did not include any form of regularization. Any results in terms of generalization are from inductive biases in gradient descent alone, e.g. the intuition discussed in Section 3.1. Regularization and noise may also play a crucial role in generalization in some settings (Zhou et al., 2019; Ziyin et al., 2025).

### 3.3. State Merger Condition

The final representational distance is exactly solved in Appendix B.2 to find:

$$||dh||^2 = \frac{1}{2}A_{\text{high}} + \sqrt{\frac{1}{4}A_{\text{high}}^2 + A_{\text{low}}^2},$$
$$(5)$$

where

$$A_{\text{high}} = ||dh(0)||^2 - \frac{N\tau_y}{\tau_h}\langle\frac{||dy_i(0)||^2}{||dh(0)||^2}\rangle_i$$
$$(6)$$
$$A_{\text{low}} = \sqrt{\frac{N\tau_y}{\tau_h}} \cdot \sqrt{\langle||y_{2,i}^* - y_{1,i}^*||^2\rangle_i}.$$

Note that these results are not dependent on the details of the neural network architecture used here. They are the results of an interaction effect locally present in any smooth, expressive machine learning system with hidden representations.

A merger occurs when the final representational distance is zero, which is when

$$A_{\text{low}} = 0 \text{ and } A_{\text{high}} < 0.$$
$$(7)$$

Suppose the network's parameters are initialized at some scale $G < 1$, where $G$ is the average decrease of representational distances when applying the recurrent map. We roughly have

$$||dh(0)||^2 \propto G^m$$
$$\langle\frac{||dy_i(0)||^2}{||dh(0)||^2}\rangle_i \propto G^n,$$
$$(8)$$

where $m = \min(m_1, m_2)$ and $n = \min(n_1, \ldots, n_N)$ are the minimal sequence lengths of the sequences corresponding to the representations and their potential subsequences respectively. Equation (7) then reduces to the condition

$$\forall_i \ y_{1,i}^* = y_{2,i}^* \text{ and } C < N \cdot G^{n-m},$$
$$(9)$$

where $C$ is an unknown constant depending on the network architecture.

We can make three observations from this:

1. The only states that can merge are those which always agree on outputs after receiving both the same sequence.

2. States only merge if the sequences they correspond to are long enough.

3. Mergers only start to occur given enough data and small enough initial weights.

We will study these observations and their implications in more detail in the next section.

# 4. Automaton Development

## 4.1. System of Interacting Particles

The interaction model requires the states to be close to each other. It does not model the global behavior of the dataset during training, only the local interaction between any two states. Many other things may occur during training, which are potentially more complex and exist on a global scale. We will ignore these effects here and treat the representational learning dynamics as a system of locally interacting particles (Liu et al., 2022; Geshkovski et al., 2023). The aim is to investigate how much of algorithm learning in recurrent neural networks can already be explained from this simple interaction alone.

## 4.2. Developing an Algorithm

We can see from Equation (9) that two states will only merge if they agree on target outputs for any possible subsequence in the data, i.e. they need not be distinguished in order to solve the task[1]. Indeed, out of all 103460 pairs of representations that ended up merging, all agreed on all possible future outputs.

Once enough of such pairs merge, the automaton will become finite. Because the merging of agreeing pairs does not affect the output of the automaton, it will still predict correctly on the training data. If, such as for the streaming parity task, the task can be expressed with a finite automaton, and the training dataset is large enough, the learned automaton and task automaton must agree on all possible sequences. In particular, as long as the training dataset contains all sequences up to the length of the task automaton's size, it is guaranteed that the automata are equivalent, as all its reachable states can be reached with the training data, on which the two automata agree.

When the learned automaton becomes finite, the behavior of the RNN becomes fixed for any sequence length, and we should see instant generalization on all lengths. This sudden complete generalization can be observed in Figure 6.

## 4.3. Redundant States

Since weights are initialized small, i.e. $G < 1$, it follows from Equation (9) that the first pairs to merge when decreasing the weight initialization are the ones for which $n - m$ is minimal. Therefore, representations corresponding to shorter sequence pairs may not merge, even when enough longer sequences do. For the training data used here, the smallest possible $n$ is 0, so we should start to see mergers once $m$ reaches a certain threshold, which can be observed in Figure 7.

---

[1]In the language of automata theory, this is equivalent to the absence of a distinguishing extension on the pair of input sequences.

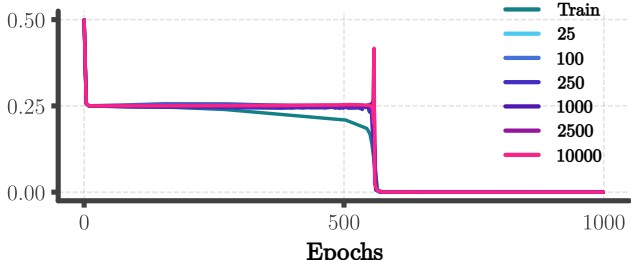

Figure 6. Validation loss for sequences of varying lengths during training on the streaming parity task. The validation loss initially does not change while the training loss goes down, but at some point it suddenly drops for all sequence lengths at the same time.

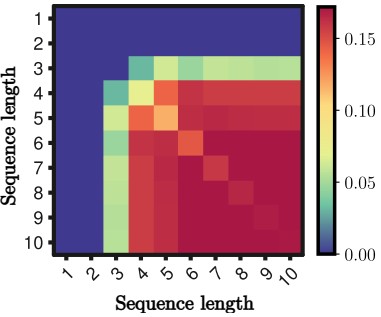

Figure 7. Fraction of pairs agreeing on all future outputs that ended up merging after training on the streaming parity task. We can see that mergers only start to occur once $m = \min(m_1, m_2)$ reaches a certain threshold.

Because not all pairs agreeing on all future outputs merge, redundant states will be present in the learned automaton. In fact, from Figure 7 it can be seen that not even all of the agreeing pairs that reached the minimum length threshold ended up merging. A possible explanation for this is that some pairs that never end up getting close during training, and therefore the effects from the interaction model do not apply.

It is not necessary for all agreeing pairs to merge to fully generalize, only enough for the final automaton to become finite. Redundant states are consequently expected to be learned. The final automaton in Figure 4 has far more states than the minimal two required to solve the task. However, it is still equivalent in computational function to the two-state automaton shown in Figure 2, which can be seen by merging all its redundant states (Figure 8).

**Redundant states in the brain** There is some evidence of redundant states in the brain (Morcos & Harvey, 2016; Marmor et al., 2023). From a functional perspective, it is not so clear why these states may exist. However, as can

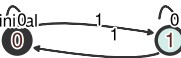

*Figure 8.* Learned automaton in the RNN from Figure 4 after redundant state reduction using Hopcroft's Algorithm (Appendix A.3). It is equivalent to the automaton representing the streaming parity task Figure 2.

be seen from the above argument, purely from a learning dynamics perspective, they are expected in the final learned automaton. Similar representations containing seemingly identical information have also been observed in other machine learning settings such as wide feed-forward networks (Doimo et al., 2021).

### 4.4. Full Generalization Phase Transition

Finally, we can see from Equation (9) that the merger condition only holds given a small enough initial weight scale $G$ and a large enough number of datapoints $N$. As either the weight initialization is decreased or the training set size is increased, agreeing states will start to merge. This can be seen on the left side of Figure 9. At some point, enough states will have merged such that the automaton becomes finite, and the RNN will generalize to all sequences. Since no state mergers can occur before the merger condition is reached, we see a sharp boundary in number of states and in particular in the validation accuracy on the right side of Figure 9, splitting the training setting landscape into two regimes: one where it fits the training data with a complete tree, and one where it learns a finite, generalizing representation of the task. Such a separation resembles the phenomenon of rich

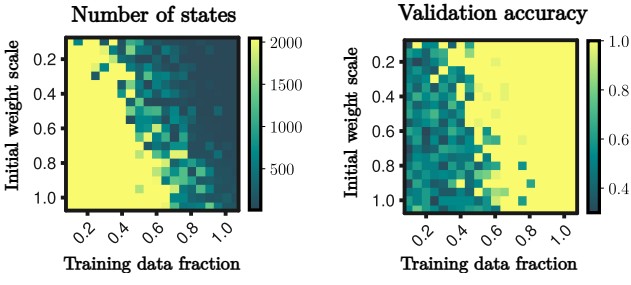

*Figure 9.* **Left**: Number of states at convergence as function of the training dataset size and initial weight scale used. **Right**: Validation accuracy of 30 sequences of length 100. All trials converged on the training set. Note the sudden jump from chance (0.5) to complete generalization (1.0) at enough data and small enough initial weights.

and lazy learning found in RNNs (Schuessler et al., 2024) and other settings (Chizat et al., 2020; Flesch et al., 2021; Atanasov et al., 2022).

## 5. Merger Dynamics

### 5.1. Two Development Phases

Something that remains unclear about the dynamics in Figure 4 is the presence of an initial phase where the automaton expands into a complete tree of all possible sequences, before a phase of mergers resulting in a finite algorithm. Why would the RNN learn to memorize individual outputs per input sequence in the first place, only to collapse into a finite automaton later?

### 5.2. Fixed Expansion Point Interaction Model

A relatively simple dynamical explanation for this behavior can be found in the representational drift of each interaction pair. If the two representations during an interaction start to drift, and their effective learning rates $1/\tau_{h_1}, 1/\tau_{h_2}$ differ, they will drift at different speeds. In this case, their distance may initially start to increase as one outpaces the other.

The local interaction model we have considered does not exhibit this behavior, as the linear expansion point is chosen to be at the moving representational mean, enforcing a representational movement symmetry. In Appendix B.3 the analytical learning trajectory for an agreeing pair is solved for in this model, and their representational distance can be seen to decay exponentially.

To allow for enough freedom in the interaction model for both representations to drift freely, we can instead keep the expansion point fixed:

$$y(x_\alpha^{(1)} \cdots x_\alpha^{(n_\alpha)} \bar{x}_i^{(1)} \cdots \bar{x}_i^{(m_i)}) \approx \bar{y}_i + D_{y_i} h_\alpha. \quad (10)$$

The downside of this choice is that as the pair drifts far away from the expansion point, the interaction model may lose accuracy. In Appendix B.4 we use a similar approach as before to reduce these dynamics to a self-contained system of 9 variables.

### 5.3. Diverging Mergers

We can see from numerical solutions to this system (Figure 10) that agreeing pairs initially diverge before they end up merging. This divergence is only present when the effective learning rates $1/\tau_{h_1}, 1/\tau_{h_2}$ differ. Experimentally, we find qualitatively similar divergence behavior in the RNN during training. We also see that the divergence occurs more often in pairs with a higher effective learning rate difference (Appendix D.3). Such an initial divergence of many agreeing pairs, may explain the tree fitting phase. A division in two phases with some similarities has been studied in feed-forward networks in (Shwartz-Ziv & Tishby, 2017).

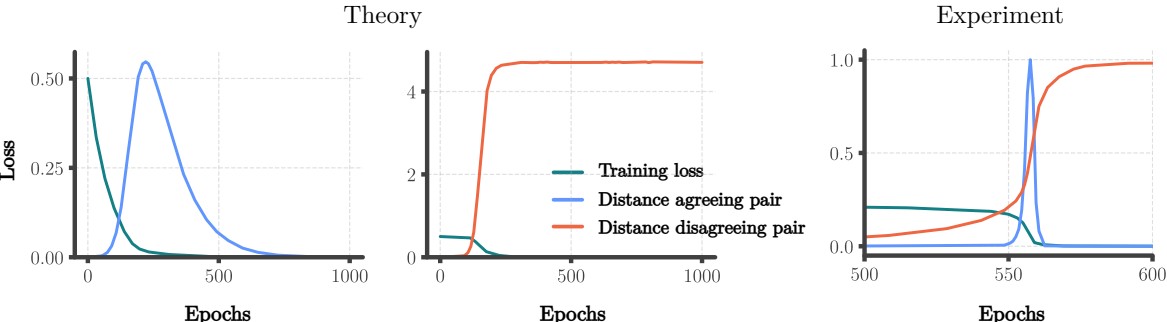

*Figure 10.* **Left**: Numerical solutions of the fixed expansion point interaction model. The representational distance of a pair agreeing on all possible future outputs first diverges before finally merging, whereas a pair which disagrees on some outputs only diverges. **Right**: Normalized experimentally observed representational distances from a randomly selected pair with agreeing outputs that ended up merging, and a randomly selected pair with disagreeing outputs.

# 6. Other Settings

## 6.1. Random Regular Tasks

The ideas considered in this paper should be applicable to any task that can be described by an automaton. Therefore, all experiments were also performed on a set of tasks defined by randomly generated automata. Similar results were found as with the streaming parity task (Appendix D.5).

## 6.2. Architecture Independence

The local interaction models discussed here are universal with respect to the neural network architecture. They represent intuitions that apply to any model with a smooth, expressive recurrent map on some hidden space and a similarly smooth and expressive output map on this space. To illustrate this, we replaced the ReLU activation function with a hyperbolic tangent and found similar qualitative results (Appendix D.6).

## 6.3. Transformers

Transformers have been shown to exhibit a similar ability to learn computational algorithms. One may wonder to what extent the ideas considered here for recurrent networks still apply to the transformer architecture.

The interpretation of the internal structure via an automaton is not as clearly applicable to a transformer. The recurrent map was an essential ingredient in the understanding of the formation of a finite automaton, as it allows for mergers to result in automaton transitions going to previous states. Interestingly enough, despite some generalization to larger sequence lengths, transformers fail to fully generalize to sequences of arbitrary length on parity computation (Anil et al., 2022).

However, the intuition behind the local interactions from continuity still applies, and so we may still find similar merg-

ing dynamics. To investigate this, we compute the number of states in a transformer during training on the modular subtraction task from (Power et al., 2022). As can be seen from Figure 11, we do not find a clear state merger pattern in the hidden representations of the transformer. However, we do see a state merger pattern in the attention matrix that is reminiscent of the two phases found in recurrent networks. A similar pattern was found for a local complexity measure in (Humayun et al., 2024).

The merging of attention patterns may possibly play an important role in out-of-distribution generalization, similar to representation merger in recurrent neural networks. Other phenomena observed here in RNNs also resemble behaviors in transformers, such as the sudden transition to full generalization (Hoffmann et al., 2024). The exact way in which a transformer can learn an algorithm from mergers is not as clear and requires further study.

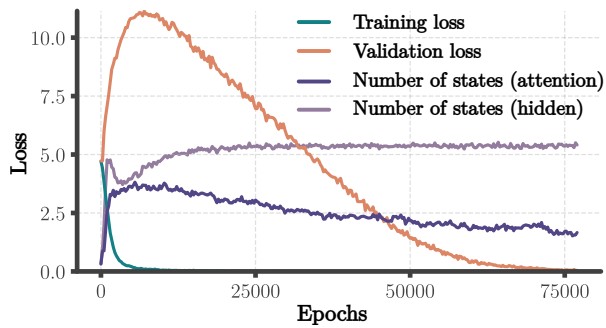

*Figure 11.* The training loss, validation loss, number of states in the attention matrix, and number of states in the hidden layer of a single layer transformer trained on modular subtraction. Note that the pattern of state mergers after initially state splitting is only present in the attention matrix, but is absent in the hidden space.

## 6.4. Discontinuity

One of the assumptions in the interaction model is continuity of the recurrent and output maps. This may not necessarily be a reasonable assumption in the context of neuroscience, as the spiking coupling between neurons is not typically viewed as continuous. For the intuitions to work, however, we only really need predictions of nearby representations to move closer when the representations do *on average*. Continuity gives us this, but may not be a necessary condition. To explore this, we add a step-continuity in the output map of the recurrent network. We find qualitatively similar results, albeit with noisier dynamics Appendix D.9.

## 7. Conclusion

Despite the surprising nature of infinite generalization from finite data, there exists a setting in which it can be understood through relatively simple intuitions about inductive bias in gradient descent. In this setting, we found that algorithm development occurs in two phases, an initial tree fitting phase and a secondary merging phase that results in generalization. The merging phase occurs via a phase transition, when the right training conditions are met. Therefore, algorithm learning and infinite generalization can occur in deep networks, but not consistently. We also saw that from a dynamical perspective, redundant states are expected in the final learned algorithm. This is of particular interest to neuroscience, as it suggests that different animals may learn different but equivalent versions of an algorithm, which is something that should be taken into account when comparing representations. Finally, we found that intuitions about automaton formation do not apply as well to transformers, and that at least for some specific tasks, recurrent networks have an advantage in terms of infinite generalization.

**Limitations** The theoretical approach used here is relatively simple and not necessarily a realistic model of the complete learning dynamics. Higher-order local interactions, global interactions, inductive biases from architectural choices, regularization, and noise were ignored, but may have additional effects on algorithm development worth studying. Additionally, the interpretation of an RNN as an automaton may provide incomplete information in more complex or continuous data settings. Other mathematical objects may be necessary to properly represent the internal structure of an RNN in such settings.

## Impact Statement

This paper presents work whose goal is to increase understanding of deep learning, which may lead to advancements in the field of Machine Learning. There are many potential societal consequences of our work, none of which we feel must be specifically highlighted here.

## Acknowledgements

We thank Stefano Sarao Mannelli and Chenxiao Ma for useful feedback. This work was supported by a Sir Henry Dale Fellowship from the Wellcome Trust and Royal Society (216386/Z/19/Z) to A.S., and the Sainsbury Wellcome Centre Core Grant from Wellcome (219627/Z/19/Z) and the Gatsby Charitable Foundation (GAT3755).

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

# A. Automaton Extraction

## A.1. Definition Deterministic Finite Automaton

Formally a deterministic finite automaton (Ashby, 1956) is a tuple $(Q, \Sigma, \delta, q_0, F)$, consisting of

1. A finite set of states $Q$

2. A finite set of possible input symbols $\Sigma$, called the alphabet

3. A transition function $\delta : Q \times \Sigma \to Q$

4. An initial state $q_0$

5. A subset of accepting states $F$

Given some string of input symbols $x = x^{(1)}x^{(2)} \dots x^{(n)}$ the automaton is said to accept the string $x$ when there exists a sequence of states $r^{(0)}, \dots, r^{(n)} \in Q$ such that

1. $r^{(0)} = q_0$

2. $\forall_i \, r^{(i+1)} = \delta(r^{(i)}, x^{(i+1)})$

3. $r^{(n)} \in F$

In the context of the streaming parity task we can take the subset of accepting states to be precisely those for which the model predicts an output 1.

## A.2. Extraction Algorithm

In order to extract one from a recurrent neural network, we define the state corresponding to an input string $x$ as the hidden representation in the network after it received that string, i.e.

$$
\begin{aligned}
q_0 &:= h_0 \\
q_{x_n \dots x_1} &:= f_h(q_{x_{n-1} \dots x_1}, x_n)
\end{aligned}. \tag{11}
$$

When two states are on top of each other, i.e. the representations are within some small distance $\epsilon$, we will consider them as the same state. Given an evaluation dataset $X$ of input sequences, the set of states can be defined as

$$
Q := \{q_x | x \in X\} / (q \sim q' \iff ||q - q'|| < \epsilon). \tag{12}
$$

The alphabet $\Sigma$ is the set of all possible input symbols in the data. The transition function $\delta(q, \sigma)$ is given by the state corresponding to $f_h(q, \sigma)$. The set of accepting states $F$ are all states for which $f_y(q)$ is closest to the output 1. If we are considering a task with more than two possible output symbols, we can generalize this definition using a Moore machine, which replaces the set of accepting states with an output function.

## A.3. State Reduction Algorithm

To help interpret the final learning automaton within the recurrent neural network, we can reduce it to a minimal state automaton with equivalent outputs. For this we use Hopcroft's Algorithm (Hopcroft, 1971) (see Algorithm 1), which returns the unique smallest DFA, equivalent to the provided DFA. Essentially what this algorithm does is it merges all pairs of states which are indistinguishable for any possible input string.

---

**Algorithm 1** Hopcroft's Algorithm

    **Input:** set of states $Q$ with output 0, set of states $F$ with output 1
    **Output:** minimal state partition $P$
    $P := \{F, Q \setminus F\}$
    $W := \{F, Q \setminus F\}$
    **while** $W$ is not empty **do**
      choose and remove a set $A$ from $W$
      **for** $c$ in $\Sigma$ **do**
        let $X$ be the set of states for which a transition on $c$ leads to a state in $A$
        **for** set $Y$ in $P$ for which $X \cap Y$ is nonempty and $Y \setminus X$ is nonempty **do**
          replace $Y$ in $P$ by the two sets $X \cap Y$ and $Y \setminus X$
          **if** $Y$ is in $W$ **then**
            replace $Y$ in $W$ by the same two sets
          **else**
            **if** $|X \cap Y| \leq |Y \setminus X|$ **then**
              add $X \cap Y$ to $W$
            **else**
              add $Y \setminus X$ to $W$
            **end if**
          **end if**
        **end for**
      **end for**
    **end while**
    **return** $P$

---

## B. Details Theoretical Analysis

### B.1. Reduction to a 3-dimensional System

To model the two-point interaction we consider two sequences $x_1^{(1)} \ldots x_1^{(m_1)}$ and $x_2^{(1)} \ldots x_2^{(m_2)}$ with nearby representations $h_1 = h(x_1^{(1)} \ldots x_1^{(m_1)})$ respectively $h_2 = h(x_2^{(1)} \ldots x_2^{(m_2)})$. Let $\mathcal{D} = \{(x_{1,i}, y_{1,i}^*), (x_{2,i}, y_{2,i}^*)\}_{i=1}^{N}$ be the set of all datapoints contained within the training dataset which have $x_1^{(1)} \ldots x_1^{(m_1)}$ or $x_2^{(1)} \ldots x_2^{(m_2)}$ as a subsequence. Assuming high expressivity, we model $h_1$ and $h_2$ as freely optimizable vectors and the output predictions of the network for each subsequent sequence in $\mathcal{D}$ as given by smooth, freely optimizable maps $y_i : H \to Y$.

In contrast to (van Rossem & Saxe, 2024), we cannot use an optimizable linearized hidden map as here we cannot smoothly vary the inputs for the hidden map. Since our input symbols are discrete, we can instead consider arbitrarily optimizable hidden vectors, as no two differing input symbols can ever get arbitrarily close to each other in the input space.

As $h_1$ and $h_2$ are close, we take a linear approximation of each output prediction map around the representational mean:

$$y(x_{\alpha,i}) = \bar{y}_i + \frac{1}{2} D_{y_i} (h_\alpha - h_{\neg\alpha}). \tag{13}$$

The mean squared loss in this approximation takes the form:

$$L = \frac{1}{2} \langle ||\bar{y}_i + \frac{1}{2} D_{y_i} (h_\alpha - h_{\neg\alpha}) - y_{\alpha,i}^*||^2 \rangle_{\mathcal{D}}. \tag{14}$$

We apply gradient decent optimization directly with respect to $D_{y_i}$, $h_\alpha$ and $\bar{y}_i$, resulting in the dynamics:

$$\frac{\mathrm{d}}{\mathrm{d}t}\bar{y}_i = -\frac{1}{\tau_{\bar{y}_i}}\frac{\partial L}{\partial \bar{y}_i}$$

$$= -\frac{1}{\tau_{\bar{y}_i}}\frac{1}{N}\langle \bar{y}_i + \frac{1}{2}D_{y_i}(h_\alpha - h_{\neg\alpha}) - y^*_{\alpha,i}\rangle_{\alpha=1,2}$$

$$= -\frac{1}{\tau_{\bar{y}_i}}\frac{1}{N}(\bar{y}_i - \frac{y^*_{2,i} + y^*_{1,i}}{2})$$

$$\frac{d}{dt}h(\alpha) = -\frac{1}{\tau_{h_\alpha}}\frac{\partial L}{\partial h_\alpha}$$

$$= -\frac{1}{\tau_{h_\alpha}}\frac{1}{4}\langle D^\top_{y_i}(\bar{y}_i + \frac{1}{2}D_{y_i}(h_\alpha - h_{\neg\alpha}) - y^*_{\alpha,i}) - D^\top_{y_i}(\bar{y}_i + \frac{1}{2}D_{y_i}(h_{\neg\alpha} - h_\alpha) - y^*_{\neg\alpha,i})\rangle_{i=1,...,N} \quad (15)$$

$$= -\frac{1}{\tau_{h_\alpha}}\frac{1}{4}\langle D^\top_{y_i}(D_{y_i}(h_\alpha - h_{\neg\alpha}) - (y^*_{\alpha,i} - y^*_{\neg\alpha,i}))\rangle_{i=1,...,N}$$

$$\frac{\mathrm{d}}{\mathrm{d}t}D_{y_i} = -\frac{1}{\tau_{y_i}}\frac{\partial L}{\partial D_{y_i}}$$

$$= -\frac{1}{\tau_{y_i}}\frac{1}{N}\frac{1}{2}\langle(\frac{1}{2}D_{y_i}(h_\alpha - h_{\neg\alpha})(h_\alpha - h_{\neg\alpha})^\top + (\bar{y}_i - y^*_{\alpha,i})(h_\alpha - h_{\neg\alpha})^\top)\rangle_{\alpha=1,2}$$

$$= -\frac{1}{\tau_{y_i}}\frac{1}{N}\frac{1}{4}(D_{y_i}(h_2 - h_1) - (y^*_{2,i} - y^*_{1,i}))(h_2 - h_1)^\top,$$

where we used the matrix differentiation identities $\frac{\partial a^\top X b}{\partial X} = ab^\top$, $\frac{\partial a^\top X^\top C X a}{\partial X} = (C + C^\top)Xaa^\top$ and $\frac{\partial ||Ax+b||^2}{\partial x} = 2A^\top(Ax + b)$.

The $\bar{y}_i$ dynamics are decoupled and can be solved directly:

$$\bar{y}_i(t) = \frac{y^*_{2,i} + y^*_{1,i}}{2} + (y_i(0) - \frac{y^*_{2,i} + y^*_{1,i}}{2})e^{-\frac{1}{\tau_{y_i}}\frac{1}{N}t}, \quad (16)$$

the solution of which takes the form of exponential decay towards each pairs target output mean.

Define $dh := h_2 - h_1$, $dy_i := D_{y_i}(h_2 - h_1)$, $w_i := ||dy_i||^2 - dy_i^\top(y^*_{2,i} - y^*_{1,i})$. We take as an Anzats representational movement of the two points towards or away from each other, i.e.

$$\frac{\mathrm{d}}{\mathrm{d}t}dh \propto dh \implies \frac{\frac{d}{dt}dh}{||\frac{d}{dt}dh||} = \frac{dh}{||dh||} \implies \frac{d}{dt}dh = \frac{||\frac{d}{dt}dh||}{||dh||}dh \quad (17)$$

Applying this twice allows us to write

$$D_{y_i}\frac{d}{dt}dh = \frac{||\frac{d}{dt}dh||}{||dh||}D_{y_i}dh = \frac{\frac{||\frac{d}{dt}dh||}{||dh||}||dh||^2}{||dh||^2}D_{y_i}dh = \frac{dh^\top(\frac{||\frac{d}{dt}dh||}{||dh||}dh)}{||dh||^2}D_{y_i}dh = \frac{dh^\top\frac{d}{dt}dh}{||dh||^2}D_{y_i}dh = \frac{1}{2}\frac{\frac{d}{dt}||dh||^2}{||dh||^2}D_{y_i}dh, \quad (18)$$

which we can use to find a self-contained scalar system:

$$
\begin{aligned}
\frac{\mathrm{d}}{\mathrm{d}t}||dh||^2 &= 2dh^\top \frac{\mathrm{d}}{\mathrm{d}t} dh \\
&= dh^\top(-\frac{1}{2}\frac{1}{\tau_h}\langle D_{y_i}^\top(D_{y_i}dh - (y_{2,i}^* - y_{1,i}^*))\rangle_{i=1,\ldots,N}) \\
&= -\frac{1}{2}\frac{1}{\tau_h}\langle||dy_i||^2 - dy_i^\top(y_{2,i}^* - y_{1,i}^*)\rangle_{i=1,\ldots,N} \\
&= -\frac{1}{2}\frac{1}{\tau_h}\langle w_i\rangle_{i=1,\ldots,N}
\end{aligned}
$$

$$
\begin{aligned}
\frac{\mathrm{d}}{\mathrm{d}t}||dy_i||^2 &= 2dy_i^\top \frac{\mathrm{d}}{\mathrm{d}t} dy_i \\
&= 2dy_i^\top(\dot{D}_{y_i}dh + \frac{dh^\top \frac{\mathrm{d}}{\mathrm{d}t}dh}{||dh||^2}D_{y_i}dh) \\
&= 2dy_i^\top(\dot{D}_{y_i}dh + \frac{1}{2}\frac{\frac{\mathrm{d}}{\mathrm{d}t}||dh||^2}{||dh||^2}D_{y_i}dh) \\
&= 2dy_i^\top(-\frac{1}{\tau_{y_i}}\frac{1}{N}\frac{1}{4}(dy_i - (y_{2,i}^* - y_{1,i}^*))||dh||^2 - \frac{1}{4}\frac{1}{\tau_h}\frac{\langle w_i\rangle_{i=1,\ldots,N}}{||dh||^2}dy_i) \\
&= -\frac{1}{\tau_{y_i}}\frac{1}{N}\frac{1}{2}(||dy_i||^2 - dy_i^\top(y_{2,i}^* - y_{1,i}^*))||dh||^2 - \frac{1}{2}\frac{1}{\tau_h}\langle w_i\rangle_{i=1,\ldots,N}\frac{||dy_i||^2}{||dh||^2}
\end{aligned}
\qquad (19)
$$

$$
\begin{aligned}
\frac{\mathrm{d}}{\mathrm{d}t}w_i &= (2dy_i - (y_{2,i}^* - y_{1,i}^*))^\top \frac{\mathrm{d}}{\mathrm{d}t}dy_i \\
&= (2dy_i - (y_{2,i}^* - y_{1,i}^*))^\top(-\frac{1}{\tau_{y_i}}\frac{1}{N}\frac{1}{4}(dy_i - (y_{2,i}^* - y_{1,i}^*))||dh||^2 - \frac{1}{4}\frac{1}{\tau_h}\frac{\langle w_i\rangle_{i=1,\ldots,N}}{||dh||^2}dy_i) \\
&= -\frac{1}{\tau_{y_i}}\frac{1}{N}\frac{1}{4}(3w_i - ||dy_i||^2 + ||y_{2,i}^* - y_{1,i}^*||^2)||dh||^2 - \frac{1}{4}\frac{1}{\tau_h}\frac{\langle w_i\rangle_{i=1,\ldots,N}}{||dh||^2}(||dy_i||^2 + w_i)
\end{aligned}
$$

where $\frac{1}{\tau_h} = \frac{1}{\tau_{h_1}} + \frac{1}{\tau_{h_2}}$.

In the case that the output effective learning rates are all equal, i.e. $\forall_i \tau_{y_i} = \tau_y$, this system can be reduced to a 3-dimensional scalar system:

$$
\begin{aligned}
\frac{\mathrm{d}}{\mathrm{d}t}||dh||^2 &= -\frac{1}{2}\frac{1}{\tau_h}\langle w_i\rangle_i \\
\frac{\mathrm{d}}{\mathrm{d}t}\langle||dy_i||^2\rangle_i &= -\frac{1}{2}(\frac{1}{N\tau_y}||dh||^2 + \frac{1}{\tau_h}\frac{\langle||dy_i||^2\rangle_i}{||dh||^2})\langle w_i\rangle_i \\
\frac{\mathrm{d}}{\mathrm{d}t}\langle w_i\rangle_i &= -\frac{1}{4}\frac{1}{N\tau_y}(3\langle w_i\rangle_i - \langle||dy_i||^2\rangle_i + \langle||y_{2,i}^* - y_{1,i}^*||^2\rangle_i)||dh||^2 - \frac{1}{4}\frac{1}{\tau_h}\frac{\langle w_i\rangle_i}{||dh||^2}(\langle||dy_i||^2\rangle_i + \langle w_i\rangle_i).
\end{aligned}
\qquad (20)
$$

## B.2. Final Representational Structure

In order to study the final representational structure learned by the network, we solve the final representational distance for the pair. Using the relationship

$$
\frac{\mathrm{d}}{\mathrm{d}t}\frac{\langle||dy_i||^2\rangle_i}{||dh||^2} = \frac{||dh||^2\frac{\mathrm{d}}{\mathrm{d}t}\langle||dy_i||^2\rangle_i - \langle||dy_i||^2\rangle_i\frac{\mathrm{d}}{\mathrm{d}t}||dh||^2}{||dh||^4} = -\frac{1}{2N\tau_y}\langle w_i\rangle_i = \frac{\tau_h}{N\tau_y}\frac{\mathrm{d}}{\mathrm{d}t}||dh||^2,
\qquad (21)
$$

we can solve $\langle||dy_i||^2\rangle_i(t)$ as a function of $||dh||^2(t)$:

$$\langle ||dy_i||^2 \rangle_i(t) = \frac{\tau_h}{N\tau_y}||dh||^4(t) + \left( \frac{\langle ||dy_i(0)||^2 \rangle_i}{||dh(0)||^2} - \frac{\tau_h}{N\tau_y}||dh(0)||^2 \right) ||dh||^2(t), \tag{22}$$

reducing the dynamics to a 2-dimensional system:

$$\frac{\mathrm{d}}{\mathrm{d}t}||dh||^2 = -\frac{1}{2}\frac{1}{\tau_h}\langle w_i \rangle_i$$

$$\frac{\mathrm{d}}{\mathrm{d}t}\langle w_i \rangle_i = -\frac{1}{4}(-\frac{\tau_h}{N^2\tau_y{}^2}||dh||^6 + \frac{1}{N\tau_y}||y_2 - y_1||^2||dh||^2 + \frac{4}{N\tau_y}||dh||^2\langle w_i \rangle_i + \frac{1}{\tau_h}\frac{\langle w_i \rangle_i^2}{||dh||^2}$$

$$+ \left( \frac{||dy(0)||^2}{||dh(0)||^2} - \frac{\tau_h}{N\tau_y}||dh(0)||^2 \right) (\frac{1}{\tau_h}\langle w_i \rangle_i - \frac{1}{N\tau_y}||dh||^4)). \tag{23}$$

This system has three fixed points

$$||dh||^2 = \frac{1}{2}A_{\text{high}} - \sqrt{\frac{1}{4}A_{\text{high}}^2 + A_{\text{low}}^2}, \langle w_i \rangle_i = 0$$

$$||dh||^2 = \frac{1}{2}A_{\text{high}} + \sqrt{\frac{1}{4}A_{\text{high}}^2 + A_{\text{low}}^2}, \langle w_i \rangle_i = 0 \text{'} \tag{24}$$

$$||dh||^2 = 0, \langle w_i \rangle_i = 0$$

where

$$A_{\text{high}} = ||dh(0)||^2 - \frac{N\tau_y}{\tau_h}\frac{\langle ||dy_i(0)||^2 \rangle_i}{||dh(0)||^2}$$

$$A_{\text{low}} = \sqrt{\frac{N\tau_y}{\tau_h}} \cdot \sqrt{\langle ||y_{2,i}^* - y_{1,i}^*||^2 \rangle_i}. \tag{25}$$

The first fixed point has negative representational distance and is thus not a valid solution.

The second fixed point has Jacobian

$$J = \begin{bmatrix} 0 & -\frac{1}{\tau_h} \\ \frac{1}{4}\frac{\tau_h}{N^2\tau_y^2}(2A_{\text{low}}^2 + \frac{1}{2}(A_{\text{high}} + \sqrt{A_{\text{high}}^2 + 4A_{\text{low}}^2})^2) & -\frac{1}{\tau_y}(\frac{1}{2}A_{\text{high}} + \sqrt{A_{\text{high}}^2 + 4A_{\text{low}}^2}) \end{bmatrix}, \tag{26}$$

with negative trace

$$\text{Tr}(J) = -\frac{1}{N\tau_y}(\frac{1}{2}A_{\text{high}} + \sqrt{A_{\text{high}}^2 + 4A_{\text{low}}^2}), \tag{27}$$

and positive determinant

$$\det(J) = \frac{1}{4}\frac{1}{N^2\tau_y^2}(2A_{\text{low}}^2 + \frac{1}{2}(A_{\text{high}} + \sqrt{A_{\text{high}}^2 + 4A_{\text{low}}^2})^2), \tag{28}$$

and is therefore always stable.

The final fixed point has Jacobian

$$J = \begin{bmatrix} 0 & -\frac{1}{\tau_h} \\ \frac{1}{2}(\frac{1}{\tau_h}\frac{\langle w_i \rangle_i^2}{||dh||^4} - \frac{1}{N\tau_y}\langle ||y_{2,i}^* - y_{1,i}^*||^2 \rangle_i) & (\frac{1}{2}\frac{1}{\tau_y}A_{\text{high}} - \frac{1}{\tau_h}\frac{\langle w_i \rangle_i}{||dh||^2}) \end{bmatrix}, \tag{29}$$

which cannot be directly evaluated at $\langle w_i \rangle_i = 0, ||dh||^2 = 0$ because of the undetermined term $\frac{\langle w_i \rangle_i}{||dh||^2}$. By replacing $\frac{\langle w_i \rangle_i}{||dh||^2}$ with the direction of approach $\frac{b}{a}$ we can solve for eigenvectors

$$\begin{bmatrix} 0 & -\frac{1}{\tau_h} \\ \frac{1}{2}(\frac{1}{\tau_h}\frac{b^2}{a^2} - \frac{1}{N\tau_y}\langle ||y_{2,i}^* - y_{1,i}^*||^2 \rangle_i) & (\frac{1}{2}\frac{1}{N\tau_y}A_{\text{high}} - \frac{1}{\tau_h}\frac{b}{a}) \end{bmatrix} \begin{bmatrix} a \\ b \end{bmatrix} = \lambda \begin{bmatrix} a \\ b \end{bmatrix} \tag{30}$$

to find

$$v_\pm = \begin{bmatrix} 1 \\ -\frac{\tau_h}{N\tau_y} \frac{A_{\text{high}} \pm \sqrt{A_{\text{high}}^2 + 4A_{\text{low}}^2}}{2} \end{bmatrix} \tag{31}$$

with one positive and one negative eigenvalue

$$\lambda_\pm = \frac{1}{N\tau_y} \frac{A_{\text{high}} \pm \sqrt{A_{\text{high}}^2 + 4A_{\text{low}}^2}}{2}. \tag{32}$$

There is always one direction along which perturbations increase, so this fixed point is not stable.

Only the second fixed point is valid and stable, hence we expect the final representational distance to reach it.

### B.3. Agreeing Pair Dynamics

When the pair agrees on all possible future outputs, i.e. $\forall_i\; y_{2,i}^* = y_{1,i}^*$ we have that $\langle w_i \rangle_i = \langle ||dy_i||^2 \rangle_i$, allowing us to reduce the system Equation (20) to

$$\frac{\mathrm{d}}{\mathrm{d}t}||dh||^2 = -\frac{1}{2}\frac{1}{\tau_h}\langle ||dy_i||^2 \rangle_i$$
$$\frac{\mathrm{d}}{\mathrm{d}t}\langle ||dy_i||^2 \rangle_i = -\frac{1}{2}\langle ||dy_i||^2 \rangle_i \left(\frac{1}{N\tau_y}||dh||^2 + \frac{1}{\tau_h}\frac{\langle ||dy_i||^2 \rangle_i}{||dh||^2}\right). \tag{33}$$

Using Equation (22) we can write a self-contained equation for $||dh||(t)$:

$$\frac{\mathrm{d}}{\mathrm{d}t}||dh||^2 = -\frac{1}{2}\frac{1}{N\tau_y}||dh||^4 + \frac{1}{2}\frac{1}{N\tau_y}A_{\text{high}}||dh||^2, \tag{34}$$

which is Bernoulli and has solution

$$||dh(t)||^2 = \frac{A_{\text{high}}}{1 + \left(\frac{A_{\text{high}}}{||dh(0)||^2} - 1\right)e^{-\frac{1}{2}\frac{1}{N\tau_y}A_{\text{high}}t}}, \tag{35}$$

which, as $A_{\text{high}} \leq ||dh(0)||^2$ exponentially decays towards the final representational distance $||dh(\infty)||^2 = A_{\text{high}}$.

### B.4. Fixed Expansion Point Interaction Model

We take the same approach before but instead keep the linear expansion point fixed during training:

$$y(x_{\alpha,i}) = b_i + D_{y_i}h_\alpha. \tag{36}$$

The mean squared loss in this approximation has the form:

$$L = \frac{1}{2}\langle ||b_i + D_{y_i}h_\alpha - y_{\alpha,i}^*||^2 \rangle_\mathcal{D}. \tag{37}$$

Motivated by the assumption of high model expressivity, we apply gradient decent optimization directly with respect to $D_{y_i}$,

$h_\alpha$ and $b_i$, resulting in the dynamics:

$$\frac{\mathrm{d}}{\mathrm{d}t}b_i = -\frac{1}{\tau_{b_i}}\frac{\partial L}{\partial b_i}$$
$$= -\frac{1}{\tau_{b_i}}\frac{1}{N}\langle b_i + D_{y_i}h_\alpha - y^*_{\alpha,i}\rangle_{\alpha=1,2}$$
$$= -\frac{1}{\tau_{\bar{y}_i}}\frac{1}{N}(b_i + D_{y_i}\langle h_\alpha\rangle_\alpha - \langle y^*_{\alpha,i}\rangle_\alpha)$$

$$\frac{d}{dt}h_\alpha = -\frac{1}{\tau_{h_\alpha}}\frac{\partial L}{\partial h_\alpha}$$
$$= -\frac{1}{\tau_{h_\alpha}}\frac{1}{2}\langle D_{y_i}^\top(b_i + D_{y_i}h_\alpha - y^*_{\alpha,i})\rangle_{i=1,\dots,N} \tag{38}$$

$$\frac{\mathrm{d}}{\mathrm{d}t}D_{y_i} = -\frac{1}{\tau_{y_i}}\frac{\partial L}{\partial D_{y_i}}$$
$$= -\frac{1}{\tau_{y_i}}\frac{1}{N}\langle D_{y_i}h_\alpha h_\alpha^\top + (b_i - y^*_{\alpha,i})h_\alpha^\top\rangle_{\alpha=1,2}$$
$$= -\frac{1}{\tau_{y_i}}\frac{1}{N}(D_{y_i}\langle h_\alpha h_\alpha^\top\rangle_{\alpha=1,2} + b_i\langle h_\alpha^\top\rangle_{\alpha=1,2} - \langle y^*_{\alpha,i}h_\alpha^\top\rangle_{\alpha=1,2}).$$

We again try the Ansatz where the representations only move towards or away from each other, which, since we take the expansion point to be the representational mean at $t = 0$, can be written by shifting coordinates without loss of generality as

$$\mathbf{h}_\alpha \propto h_\alpha\mathbf{v}, \tag{39}$$

for some vector $\mathbf{v}$ with $||\mathbf{v}|| = 1$. We define $d_i := \mathbf{v}^\top D_{y_i}\mathbf{v}$, $b_i := \mathbf{v}^\top D_{y_i}^\top \mathbf{b}_i$, allowing us write using the derivatives

$$\frac{d}{dt}h_\alpha = -\frac{1}{\tau_{h_\alpha}}\frac{1}{2}\langle(D_{y_i}v)^\top(b_i + h_\alpha D_{y_i}v - y^*_{\alpha,i})\rangle_i$$
$$\frac{\mathrm{d}}{\mathrm{d}t}b_i = -\frac{1}{\tau_{b_i}}\frac{1}{N}(b_i + \langle h_\alpha\rangle_\alpha D_{y_i}v - \langle y^*_{\alpha,i}\rangle_\alpha) \tag{40}$$
$$\frac{\mathrm{d}}{\mathrm{d}t}D_{y_i}v = -\frac{1}{\tau_{y_i}}\frac{1}{N}(\langle h_\alpha^2\rangle_\alpha D_{y_i}v + \langle h_\alpha\rangle_\alpha b_i - \langle h_\alpha y^*_{\alpha,i}\rangle_\alpha),$$

a scalar system which takes the form

$$\frac{d}{dt}h_\alpha = -\frac{1}{\tau_{h_\alpha}}\frac{1}{2}\langle b_i^\top D_{y_i}v + h_\alpha||D_{y_i}v||^2 - y^\top_{\alpha,i}D_{y_i}v\rangle_i$$

$$\frac{\mathrm{d}}{\mathrm{d}t}b_i^\top D_{y_i}v = -\frac{1}{\tau_{b_i}}\frac{1}{N}(b_i^\top D_{y_i}v + \langle h_\alpha\rangle_\alpha||D_{y_i}v||^2 - \langle y^\top_{\alpha,i}D_{y_i}v\rangle_\alpha) - \frac{1}{\tau_{y_i}}\frac{1}{N}(\langle h_\alpha^2\rangle_\alpha b_i^\top D_{y_i}v + \langle h_\alpha\rangle_\alpha||b_i||^2 - \langle h_\alpha b_i^\top y^*_{\alpha,i}\rangle_\alpha)$$

$$\frac{\mathrm{d}}{\mathrm{d}t}b_i^\top y^*_{\beta,i} = -\frac{1}{\tau_{b_i}}\frac{1}{N}(b_i^\top y^*_{\beta,i} + \langle h_\alpha\rangle_\alpha y^\top_{\beta,i}D_{y_i}v - y^\top_{\beta,i}\langle y^*_{\alpha,i}\rangle_\alpha)$$

$$\frac{\mathrm{d}}{\mathrm{d}t}||b_i||^2 = -2\frac{1}{\tau_{b_i}}\frac{1}{N}(||b_i||^2 + \langle h_\alpha\rangle_\alpha b_i^\top D_{y_i}v - \langle b_i^\top y^*_{\alpha,i}\rangle_\alpha)$$

$$\frac{\mathrm{d}}{\mathrm{d}t}||D_{y_i}v||^2 = -2\frac{1}{\tau_{y_i}}\frac{1}{N}(\langle h_\alpha^2\rangle_\alpha||D_{y_i}v||^2 + \langle h_\alpha\rangle_\alpha b_i^\top D_{y_i}v - \langle h_\alpha y^\top_{\alpha,i}D_{y_i}v\rangle_\alpha)$$

$$\frac{\mathrm{d}}{\mathrm{d}t}y^{*\top}_{\beta,i}D_{y_i}v = -\frac{1}{\tau_{y_i}}\frac{1}{N}(\langle h_\alpha^2\rangle_\alpha y^{*\top}_{\beta,i}D_{y_i}v + \langle h_\alpha\rangle_\alpha y^{*\top}_{\beta,i}b_i - \langle h_\alpha y^{*\top}_{\beta,i}y^*_{\alpha,i}\rangle_\alpha)$$

$$\tag{41}$$

If the output effective learning rates are again all equal, i.e. $\forall\ \tau_{b_i} = \tau_b, \forall\ \tau_{y_i} = \tau_y$, this system can be reduced to 9 scalars:

$$
\frac{d}{dt}h_\alpha = -\frac{1}{\tau_{h_\alpha}}\frac{1}{2}(\langle b_i^\top D_{y_i} v\rangle_i + h_\alpha\langle ||D_{y_i}v||^2\rangle_i - \langle y_{\alpha,i}^\top D_{y_i}v\rangle_i)
$$

$$
\frac{\mathrm{d}}{\mathrm{d}t}\langle b_i^\top D_{y_i}v\rangle_i = -\frac{1}{\tau_b}\frac{1}{N}(\langle b_i^\top D_{y_i}v\rangle_i + \langle h_\alpha\rangle_\alpha\langle ||D_{y_i}v||^2\rangle_i - \langle y_{\alpha,i}^\top D_{y_i}v\rangle_{\alpha,i})
$$

$$
-\frac{1}{\tau_y}\frac{1}{N}(\langle h_\alpha^2\rangle_\alpha\langle b_i^\top D_{y_i}v\rangle_i + \langle h_\alpha\rangle_\alpha\langle ||b_i||^2\rangle_i - \langle h_\alpha\langle b_i^\top y_{\alpha,i}^*\rangle_i\rangle_\alpha)
$$

$$
\frac{\mathrm{d}}{\mathrm{d}t}\langle b_i^\top y_{\beta,i}^*\rangle_i = -\frac{1}{\tau_b}\frac{1}{N}(\langle b_i^\top y_{\beta,i}^*\rangle_i + \langle h_\alpha\rangle_\alpha\langle y_{\beta,i}^\top D_{y_i}v\rangle_i - \langle y_{\beta,i}^\top y_{\alpha,i}^*\rangle_{\alpha,i}) \tag{42}
$$

$$
\frac{\mathrm{d}}{\mathrm{d}t}\langle ||b_i||^2\rangle_i = -2\frac{1}{\tau_b}\frac{1}{N}(\langle ||b_i||^2\rangle_i + \langle h_\alpha\rangle_\alpha\langle b_i^\top D_{y_i}v\rangle_i - \langle b_i^\top y_{\alpha,i}^*\rangle_{\alpha,i})
$$

$$
\frac{\mathrm{d}}{\mathrm{d}t}\langle ||D_{y_i}v||^2\rangle_i = -2\frac{1}{\tau_y}\frac{1}{N}(\langle h_\alpha^2\rangle_\alpha\langle ||D_{y_i}v||^2\rangle_i + \langle h_\alpha\rangle_\alpha\langle b_i^\top D_{y_i}v\rangle_i - \langle h_\alpha\langle y_{\alpha,i}^\top D_{y_i}v\rangle_i\rangle_\alpha)
$$

$$
\frac{\mathrm{d}}{\mathrm{d}t}\langle y^{*\top}_{\beta,i}D_{y_i}v\rangle_i = -\frac{1}{\tau_y}\frac{1}{N}(\langle h_\alpha^2\rangle_\alpha\langle y^{*\top}_{\beta,i}D_{y_i}v\rangle_i + \langle h_\alpha\rangle_\alpha\langle y^{*\top}_{\beta,i}b_i\rangle_i - \langle h_\alpha\langle y^{*\top}_{\beta,i}y_{\alpha,i}^*\rangle_i\rangle_\alpha)
$$

**Training loss** The loss can be written expressed using the variables from this system

$$
L = \frac{1}{2}(\langle ||b_i||^2\rangle_i + \langle h_\alpha^2\rangle_\alpha\langle ||D_{y_i}v||^2\rangle_i + \langle ||y_{\alpha,i}^*||^2\rangle_{\alpha,i} + 2\langle h_\alpha\rangle_\alpha\langle b_i^\top D_{y_i}v\rangle_i - 2\langle h_\alpha\langle y^{*\top}_{\alpha,i}D_{y_i}v\rangle_i\rangle_\alpha - 2\langle\langle y^{*\top}_{\alpha,i}b_i\rangle_i\rangle_\alpha). \tag{43}
$$

**Equal hidden effective learning rates** When the pairs agree on all outputs,s we have

$$
\langle y^{*\top}_{1,i}y_{\alpha,i}^*\rangle_{\alpha,i} = \langle y^{*\top}_{2,i}y_{\alpha,i}^*\rangle_{\alpha,i} \tag{44}
$$

Assuming no correlation at initialization due to random weights

$$
\langle b_i^\top D_{y_i}v\rangle_i(0) = 0
$$
$$
\langle b_i^\top y_{\beta,i}^*\rangle_i(0) = 0, \tag{45}
$$
$$
\langle y^{*\top}_{\beta,i}D_{y_i}v\rangle_i(0) = 0
$$

we find the relations

$$
\frac{\mathrm{d}}{\mathrm{d}t}\langle b_i^\top y_{1,i}^*\rangle_i = \frac{\mathrm{d}}{\mathrm{d}t}\langle b_i^\top y_{2,i}^*\rangle_i
$$
$$
\frac{\mathrm{d}}{\mathrm{d}t}\langle y^{*\top}_{1,i}D_{y_i}v\rangle_i = \frac{\mathrm{d}}{\mathrm{d}t}\langle y^{*\top}_{2,i}D_{y_i}v\rangle_i \tag{46}
$$

such that

$$
\langle b_i^\top y_{1,i}^*\rangle_i = \langle b_i^\top y_{2,i}^*\rangle_i
$$
$$
\langle y^{*\top}_{1,i}D_{y_i}v\rangle_i = \langle y^{*\top}_{2,i}D_{y_i}v\rangle_i \tag{47}
$$

From this it follows that

$$
\frac{d}{dt}(\tau_2 h_2 - \tau_1 h_1) = -\frac{1}{2}((h_2 - h_1)\langle ||D_{y_i}v||^2\rangle_i). \tag{48}
$$

In the case of equal effective hidden learning rates $\tau_{h_1} = \tau_{h_2}$, the representational distance can only increase

$$
\frac{d}{dt}||h_2 - h_1||^2 = -\langle ||D_{y_i}v||^2\rangle_i, \tag{49}
$$

so no initial divergence occurs for agreeing pairs in this case.

# C. Experimental Details

For all experiments, the PyTorch library was used to train the models. The code we used can be found at `https://github.com/loekvanrossem/rnn_structure.git`.

## C.1. Streaming Parity Task

**Model** The neural network architecture consists of a single fully connected recurrent layer with 100 hidden units, ReLU activation function, along with a fully connected linear output layer, both with bias:

$$h_t = \text{ReLU}(x_t W_{ih}^\top + b_{ih} + h_{t-1} W_{hh}^\top + b_{hh})$$
$$y_t = W_{hy}^\top h_t + b_{hy}$$
(50)

The initial hidden vector $h_0$ was set to all ones, to reduce time being stuck at the zero fixed point early in training.

**Training procedure** The model was trained on all sequences up to length 10. We used 100 randomly selected sequences of length 50 to compute the validation loss during training. Both the inputs and outputs were embedded in a 2-dimensional space using a one-hot encoding.

Optimization was performed using stochastic gradient descent on the mean squared training loss. The model was trained over 1000 epochs at a learning rate of 0.02 and a batch size of 128. No momentum or regularization was used. Weights were initialized small using Xavier initialization at a gain of 0.1. Bias was initialized at zero.

**Automata** Automata were extracted using the complete training set. The merger cutoff was set at a factor of 0.01 of the representational standard deviation.

## C.2. Grokking

For the re-implementation of the grokking experiment, we made use of the following publicly available repository: `https://github.com/Sea-Snell/grokking.git`

The data used is subtraction modulo 96, with a fraction of 0.6 of possible examples set aside for validation.

Every 250 epochs we evaluated the training loss, validation loss, number of states in the attention, and number of states in the hidden layer.

To compute the number of states in the hidden activations, we considered the activations after the normalization layer, as suggested in (Adriaensen & Maene, 2024). Next, we applied a principal component analysis to reduce the space to 512 dimensions and merged pairs with distance below a threshold to count the number of states. The number of states in the attention matrix was computed in a similar manner.

The threshold was chosen to be 1.25 times the initial maximum distance, as states are expected to be close at initialization. Any significantly larger or smaller choice for the threshold resulted in the counting of a single or all possible states at every epoch.

# D. Supplemental Figures

## D.1. No Representational Contraction

One may wonder if the reduction in the number of automaton states during the second phase of learning is merely the result of the whole representational space contracting, as the merger threshold is a fixed constant. To make sure this is not the case, we computed the representational mean distance over time and found it always increases (Figure 12).

## D.2. Validity Linear Approximation

The theoretical interaction model assumes close enough representations such that the first order term in the Taylor expansion dominates higher order terms. One may wonder whether or not representations in practice are ever close enough for this to be a reasonable assumption, especially since even pairs that end up merging still initially move apart from each other as can be seen in Figure 10. To test this we compute the average ratio of the norm of the second-order Taylor term with respect to

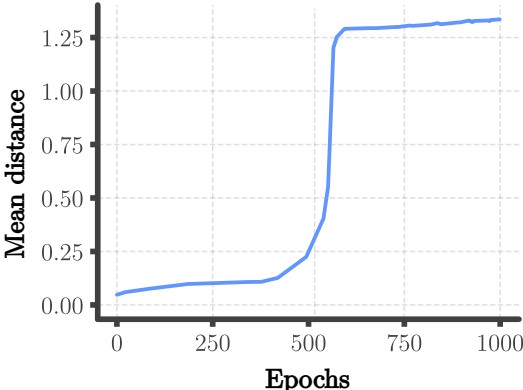

*Figure 12.* The mean pairwise representational distance of training set during training on the streaming parity task. It never decreases during training, hence state mergers cannot be entirely explained by representational contraction.

the first-order term (Figure 13). We find that although the linear approximation does get worse when representations start to move apart, the first-order term still remains dominant, and in the case of merging pairs by at least a factor of 10.

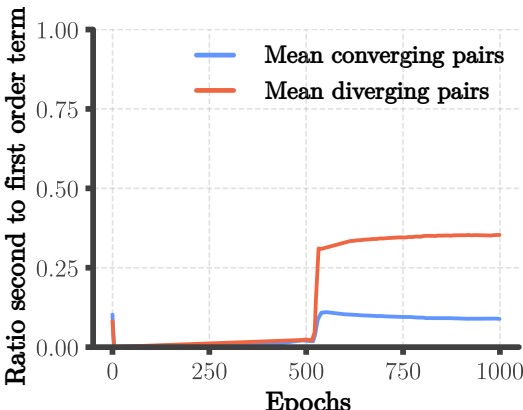

*Figure 13.* The norms of the second-order term divided by the first-order term in the Taylor expansion of the output map during training, averaged over all pairs that ended up below the merging threshold and all pairs that ended up above the merging threshold out of 100 randomly selected pairs. Note that this ratio increases around the point where pairs start to diverge, however, the first order term remains dominant on average throughout the entire training procedure.

### D.3. Diverging Mergers Occurrence Rates

As gradient descent scales with the number of parameters, the effective learning rates of representations should be proportional to the sequence lengths.

$$\frac{1}{\tau_h} \propto n \tag{51}$$

This is because for a hidden state corresponding to a sequence of length $n$, the map assigning that hidden state consists of the recurrent map applied $n$ times. Therefore, this map has $nP$ parameters, where $P$ is the number of parameters on the recurrent map, and parameters that appear multiple times are double counted. Applying gradient descent will thus result in a sum of $nP$ terms, and this increases linearly with $n$.

From Equation (51) we would expect such mergers that initially diverge to be more prevalent among pairs with a large difference in sequence lengths. Such a pattern was indeed observed in the experiment (Figure 14).

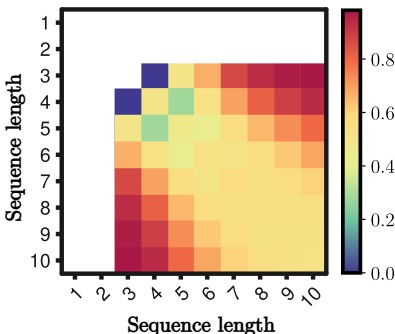

*Figure 14.* Fraction of mergers that first move apart by least a factor 10 of the merging threshold, at some point during training, before merging in the end. Displayed as a function of the sequence lengths of the points in each pair. We can see that pairs with a larger difference in sequence length appear to diverge initially more often during merging.

### D.4. State Change Patterns Due to Diverging Merger

According to the theory, the reason the number of states initially increases before decreasing is that pairs of datapoints agreeing on all outputs will first move apart before they end up merging. As we can see Figure 15, the decrease of the number of states after the initial increase can indeed be entirely explained by merging pairs that initially diverged.

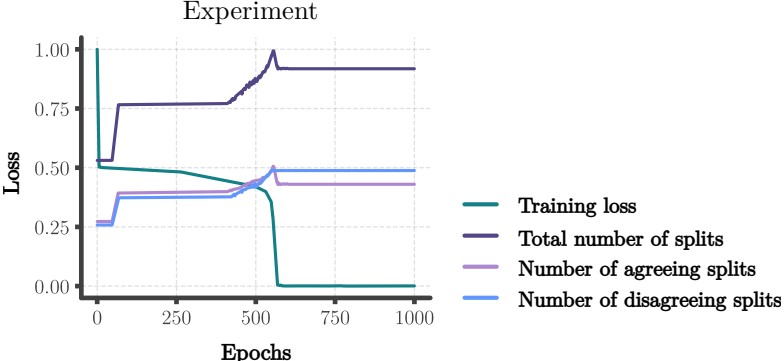

*Figure 15.* The number of pairs of datapoints whose representational distance are apart more than the merging threshold during training on the streaming parity task, divided by whether or not they agree on all possible future outputs. We see that the drop in number of automaton states can be explained by pairs agreeing on all outputs initially diverging before they end up merging.

### D.5. Random Regular Tasks

We ran multiple experiments on randomly generated regular tasks with a number of states between 1 and 7. Tasks were defined using random automata, which were generated by starting with an initial state and adding transitions for each input symbol to either a new state with some probability which we set to 0.75, or a uniformly selected already existing state. This procedure was iterated for each state until all transitions were assigned, or 7 states exist at which point all remaining transitions would be assigned to already existing states. Results were found to be consistently qualitatively similar to the streaming parity task. Initial weight gain was set to 0.025, the learning rate to 0.05. An example is shown in Figure 16. Other samples of regular tasks provided qualitatively similar results.

### D.6. Hyperbolic Tangent Activation Function

We reran the experiments on the same recurrent neural network architecture except with hyperbolic tangent non-linearity. Initial weight gain was set to 0.01 and learning rate to 0.05. All other settings remained the same. Similar qualitative results

were found as with the ReLU network (Figure 17).

### D.7. Modular Addition

The single-layer transformer trained on modular subtraction has a much smoother transition to generalization than the RNN trained on parity. Modular addition tends to have a sharper transition. To see if this changes any of the results we run the experiment again. We still find similar merging patterns (Figure 18).

### D.8. State Merging in the Value Matrix

Although less clear, we found something resembling a state merging effect in the value matrix of the transformer (Figure 19). The key matrix did not show such an effect.

### D.9. Discontinuity

We can break the assumption of continuity by adding a step discontinuity in the output map of the RNN.

$$f(x) = \begin{cases} 0 & \text{if } x < 0 \\ x + 1 & \text{if } x > 0 \end{cases} \tag{52}$$

We still find similar results albeit with nosier dynamics (Figure 20).

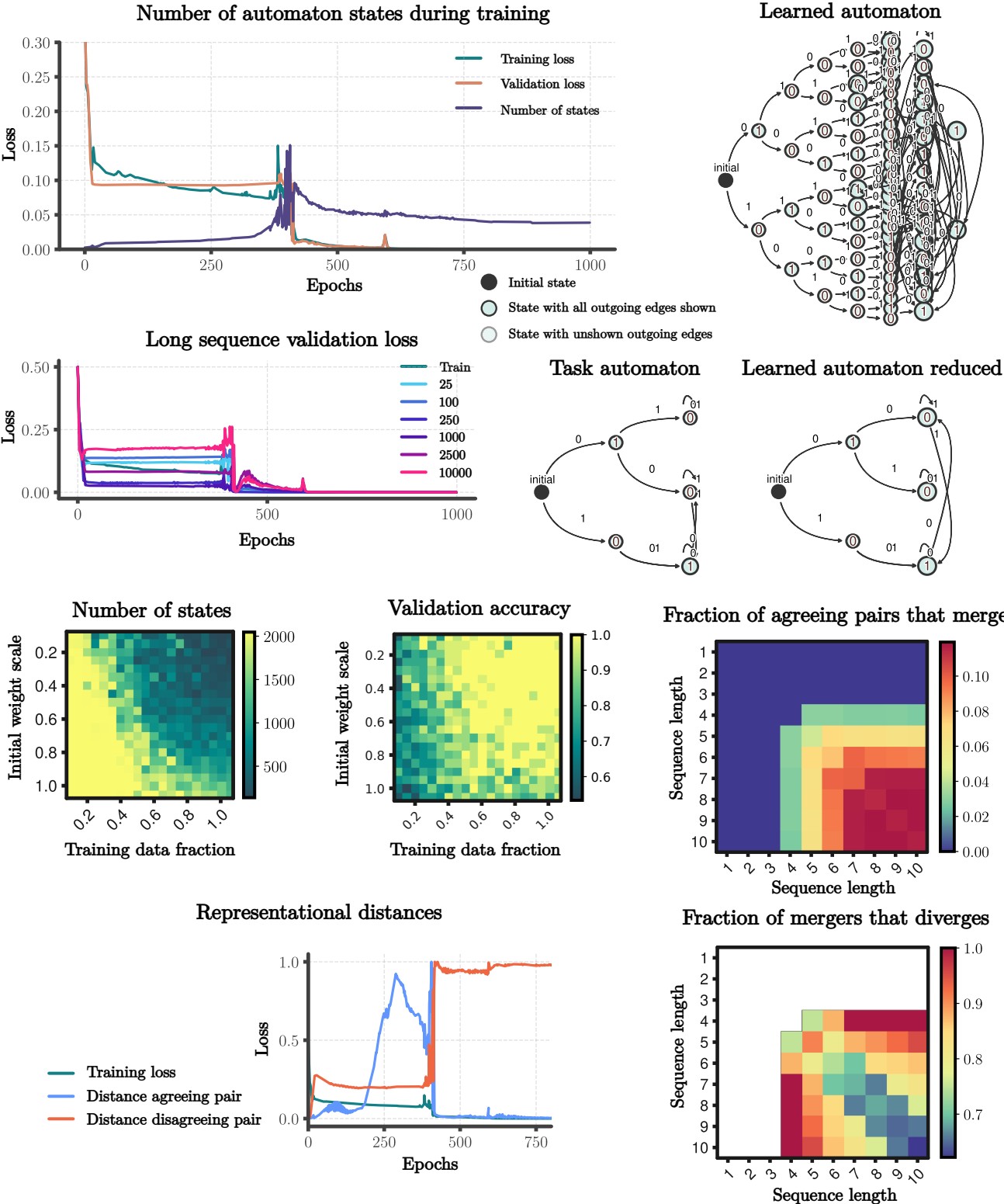

Figure 16. Experimental results of a recurrent neural network trained on a randomly generated regular task. Note that the task automaton and learned automaton are equal, but states are displayed in different locations.

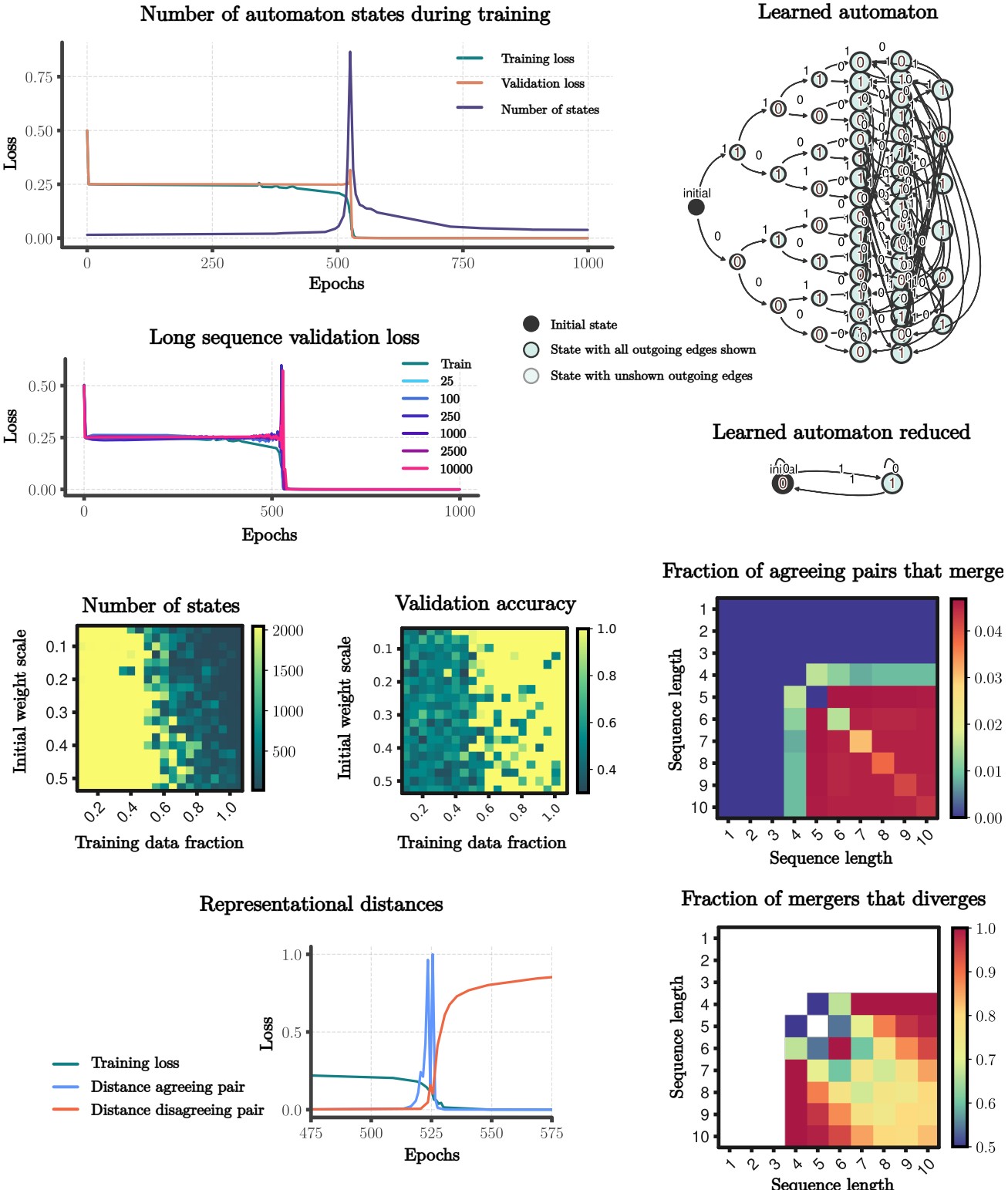

*Figure 17.* Experimental results of a recurrent neural network with hyperbolic tangent non-linearity trained on the streaming parity task.

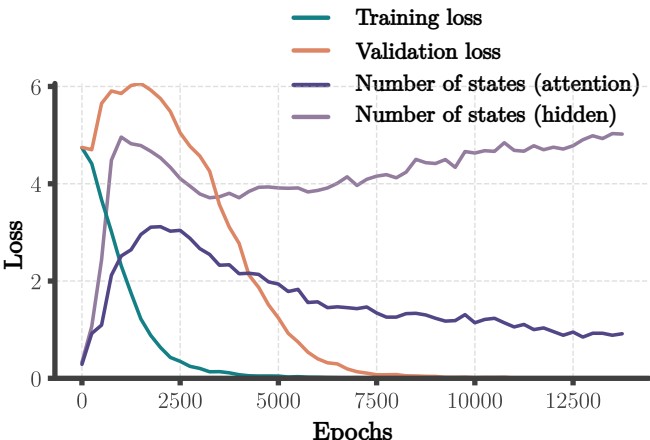

*Figure 18.* The training loss, validation loss, number of states in the attention matrix, and number of states in the hidden layer of a single layer transformer trained on modular addition. Note that as with subtraction we see a pattern of initial attention pattern divergence followed by mergers, while hidden states continue to increase.

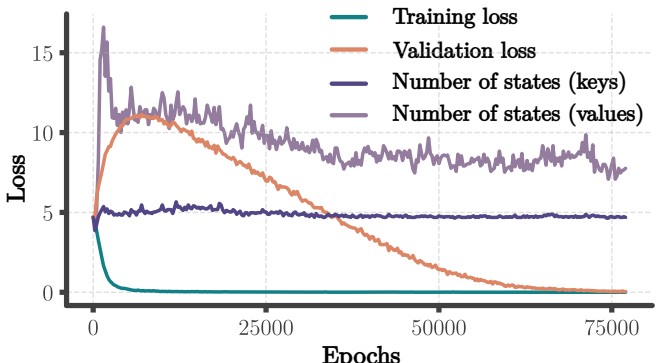

*Figure 19.* The training loss, validation loss, number of states in the key matrix, and number of states in the value matrix of a single layer transformer trained on modular subtraction. Note that some amount of state merging appears to happen in the value matrix, but not for the key matrix. Losses have been scaled down for visualization purposes.

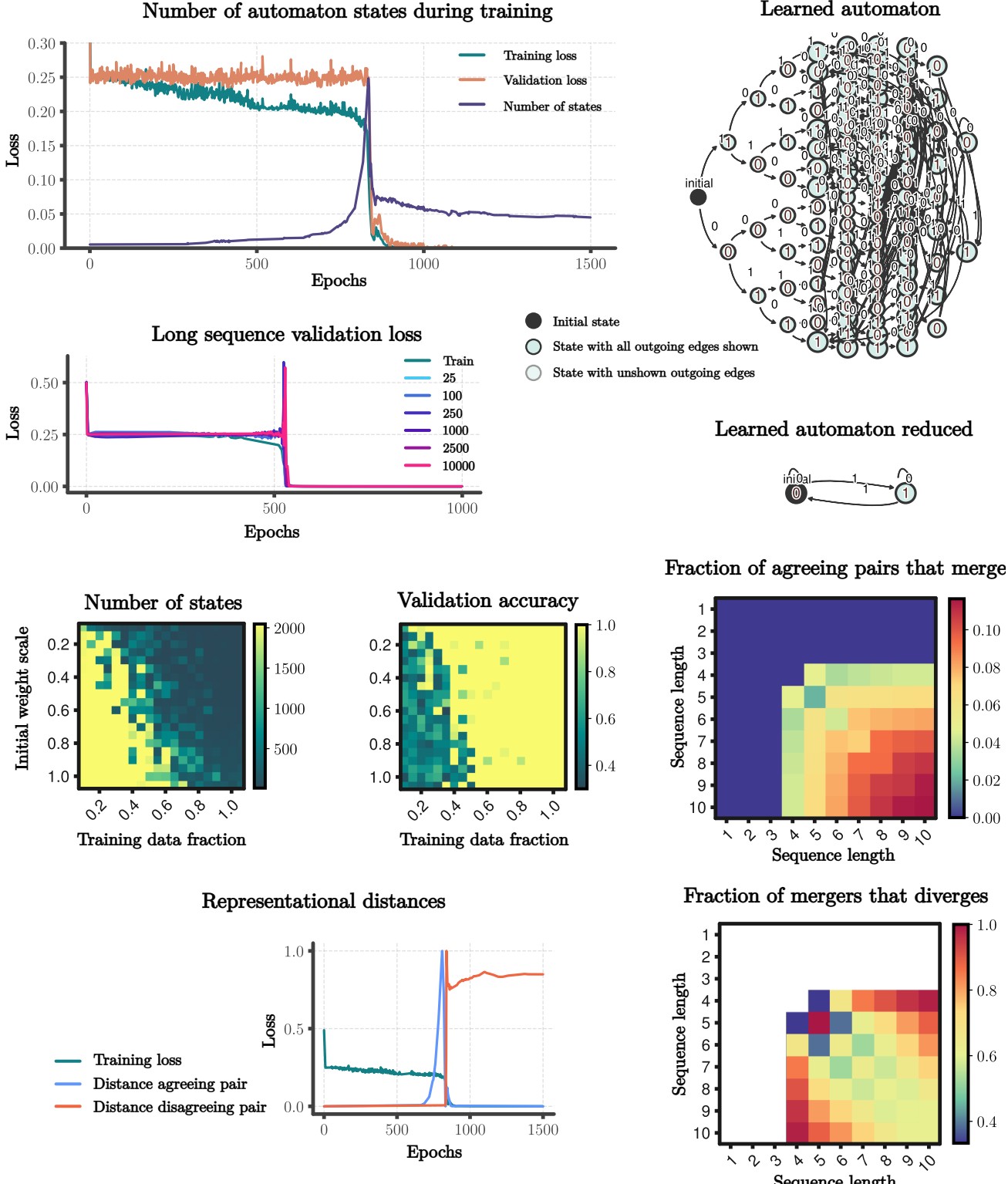

*Figure 20.* Experimental results of a recurrent neural network with a discontinuity in the output map trained on the streaming parity task.

