# OpenReview forum: "Algorithm Development in Neural Networks: Insights from the Streaming Parity Task"
_ICML.cc/2025/Conference — ICML 2025 oral_

### Official Review · Reviewer_axEp · 2025-03-11

**Overall Recommendation:** 4

**Summary:**

This is an interesting work that studies the development of algorithms in RNN. It combines theory and experiments, which I think is a great plus. The theory and experiments are novel, showing how representations become merged in a linearized dynamical theory

**Claims And Evidence:**

The claims are validated, and I find that the experiments are sufficient

**Essential References Not Discussed:**

I think the references are fine -- maybe the work should acknowledge and discuss its limitation more. For example, noise and/or regularization are known to be a key factor of representation learning, but the proposed theory ignores both factors: https://arxiv.org/abs/2410.03006

**Experimental Designs Or Analyses:**

I am a little confused with the result in Figure 4.
a) at initialization, why are there so few states? Here, all the weights are random and so all the states should be different, right? I would expect a lot of states here

b) how do you count the number of states? Does it require both representations to be **exactly** identical? Or some approximate identicality is sufficient?

**Methods And Evaluation Criteria:**

Good

**Other Comments Or Suggestions:**

NA

**Other Strengths And Weaknesses:**

NA

**Questions For Authors:**

See my answers above

**Relation To Broader Scientific Literature:**

NA

**Theoretical Claims:**

Nothing too problematic, even though the theory may be too simple -- it is a simple quadratic loss function!

---

> ### Author Rebuttal · Authors · 2025-03-31
>
> Thank you for the feedback and spending your time reviewing our paper. Please find below responses to your comments and the changes we will make to the paper.
>
> ***Experimental Designs Or Analyses***
>
> - The RNN was initialized at small weights, which is why we see few states at initialization in Figure 4. Due to the small weights, the hidden states are initially all close. We will make this clearer in the figure's description. When we rerun the experiment at a higher initialization scale (but still in the generalizing regime), we see more states at initialization. These quickly disappear during training, and the rest of the behavior is the same in this case as seen in Figure 4.
>
> - We used approximate identity, since due to numerics two representations never end up overlapping exactly. We merge two states when their representational distance is below a threshold. The threshold was set at 0.01 of the representational standard deviation during the entire training procedure. Varying the threshold around this scale did not significantly affect results.
>
> ***Essential References Not Discussed***
>
> We agree the paper should make it more clear that noise and regularization can still play important part in representation learning for many settings. We will add this to the paper and cite relevant literature.

---

### Official Review · Reviewer_opdC · 2025-03-12

**Overall Recommendation:** 4

**Summary:**

The authors provide an in-depth analysis for an RNN solving the Streaming Parity Task. Specifically, they extract a computational graph from the network at different training phases. Once this graph becomes cyclic, the network can generalize to longer times. An analytical treatment of how learning dynamics affects nearby representations explains some of the observed properties

## After rebuttal

After reading all the rebuttals and discussion with all reviewers, I am keeping my score.

**Claims And Evidence:**

Overall yes. There is one point which seems at odds with the results. In line 155, the authors write “As the model trains, the automaton expands into a complete tree that correctly fits the training data, reducing the loss only on the training data. After that, states in the automaton appear to merge until it becomes finite, at which point it generalizes on all data.”
From reading this, one would expect the training loss to be close to zero – while the states keep on merging and then there is a large drop in generalization without a big change in training loss. In contrast, Figure 6 shows that the training loss hardly changes until dropping almost together with the generalization loss.

**Essential References Not Discussed:**

Two papers that come to mind in the context of automaton extraction are:
Turner, Elia, Kabir V Dabholkar, and Omri Barak. “Charting and Navigating the Space of Solutions for Recurrent Neural Networks.” In Advances in Neural Information Processing Systems, 34:25320–33. Curran Associates, Inc., 2021.
Brennan, Connor, Adeeti Aggarwal, Rui Pei, David Sussillo, and Alex Proekt. “One Dimensional Approximations of Neuronal Dynamics Reveal Computational Strategy.” PLOS Computational Biology 19, no. 1 (January 6, 2023): e1010784. https://doi.org/10.1371/journal.pcbi.1010784.

**Experimental Designs Or Analyses:**

Yes. The task is well suited to the problem at hand, and the training and automaton extraction seem valid.

**Methods And Evaluation Criteria:**

Yes. The tasks chosen are all described by automatons, allowing to check whether state mergers can lead to finite automatons and thereby generalization.

**Other Comments Or Suggestions:**

None.

**Other Strengths And Weaknesses:**

Finite automaton (line 160) – because the network is finite, there is a finite number of states by definition. To show that the automaton is infinite before the transition, one would need to see how the number of states scales with the size of the network and the duration of the training sequences.
What determines the timescales (tau_h, tau_y)?
Definition of w is a bit confusing because it seems like dy should equal y2-y1. Perhaps it’s worth emphasizing that dy is the network’s output and y2-y1 is the difference in labels.
Line 1175 wether – whether
Figure 14 – The match between theory and experiment is not very convincing.
Relation to neural collapse, Farrell. NTK and others. Tunnel

Line 194 – “this assumption is true for the training data”. What happens when x1,x2 are of different length? Because the training set is length 10, there isn’t a match between any residual sequences.
The definition of subsequence vs. initial sequence is not entirely clear.
Equation 49 – what is the intuition / justification for this?
There is no direct comparison of dynamics (dh,w…dy) to data. (Similar to figure 4 in van Rossem & Saxe).
Figure 4 :loss type and secondary y axis missing

**Questions For Authors:**

None.

**Relation To Broader Scientific Literature:**

The mathematical part is very similar to van Rossem & Saxe 2024. If I understand correctly, the main difference is defining h_1 and h_2 as individual functions, instead of optimizing Dh. This choice should be explicitly stated and motivated in the text.
The application of the theory to algorithm development is novel to the best of my knowledge.

**Theoretical Claims:**

I read all the proofs. Did not check every math step in detail. The methods are very similar to van Rossem & Saxe 2024.

---

> ### Author Rebuttal · Authors · 2025-03-31
>
> Thank you for the feedback and spending your time reviewing our paper. Please find below responses to your comments and the changes we will make to the paper.
>
> ***Claims And Evidence***
>
> The phrasing of the sentence in line 155 is not clear enough, so we will amend it to avoid confusion. What is meant by this is that initially only the training loss decreases, even though the generalization loss remains fixed. We can see in Figure 6 that the generalization loss is completely fixed until around 550 epochs, whereas training loss already can be seen in the plot to be decreasing at around 250 epochs, albeit very slowly initially. By the time the training loss gets close to zero the generalization loss starts to drop, although this happens almost immediately, with no delay in between.
>
> ***Relation To Broader Scientific Literature***
>
> Correct, the only difference between the interaction model discussed here in 3.2 and that in van Rossem & Saxe 2024 is replacing the linearized representation map with two optimizable vectors. This difference is due to the consideration of an RNN as opposed to a feedforward network. In a feedforward network we can smoothly vary the inputs for the map that assigns representations, and can thus take a local approximation. For an RNN we cannot do this as easily as the hidden map will depend on multiple input symbols. But since these are discrete we can instead consider representations for different input sequences as each separately optimizable vectors, as different sequences cannot get arbitrarily close in the input space. We will mention this contrast explicitly in the text.
>
> ***Other Strengths And Weaknesses***
>
> Note that the network used here is quite overparameterized compared to the task. The RNN has 100 hidden units, which is very high dimensional considering a representation solving the task can in theory be constructed in one dimension. We did not see any issues related to network size limitations in the setting we studied. We found that we got consistent results for finiteness of the automata, using a large enough test set, with long enough sequences.
>
> The timescales $\tau_h$, $\tau_y$ are depend on the architectural details of the map assigning representations and the map assigning outputs to those representations respectively, which have been abstracted away in the theoretical model. They cannot be determined from theory, only empirically by studying trajectories.
>
> We will change the notation to more clearly distinguish between predicted outputs and target outputs.
>
> In Figure 14 we see that for both theory and experiment the drop in number of states in the second phase is solely due to agreeing pairs. This is the main point of the figure. Besides that there is indeed not much of a match, and it the figure does not explain much else. We will state this more clearly in the paper.
>
> The assumption in line 194 indeed does not hold in the training data for sequence pairs with different lengths and where only one subsequence will result in a sequence longer than 10. It still holds for most pairs of two input sequences and subsequence, but we should not say it holds for all. The better motivation for the assumption is that when there isn't a matching sequence that pair will not contribute to the interaction. We will change this in the text.
>
> The motivation for equation (49) is as follows: if we consider the map assigning a representation to a sequence of length $n$, this map consists of the recurrent map applied $n$ times. Suppose this recurrent map has $P$ parameters. The total map assigning the representation will then have $nP$ parameters, if we double count the same parameters when they show up multiple times. Applying gradient descent to this will result in a sum of $nP$ terms, thus we expect the effective learning rate to scale linearly with $n$. We will add this explanation to the paper, since the current one is too limited.
>
> There is no comparison of the two point dynamics because the simpler model from van Rossem & Saxe 2024 was not enough to capture the behavior here, see e.g. the initial divergence in Figure 10. The fixed expansion point interaction model can capture this behavior, but some of its parameters are hard to measure, as some of the quantities in equation (41) cannot be directly computed from the representation and prediction vectors. Additionally, it has far more freedom considering it has 4 instead of 2 effective parameters. Fitting it to the data would not be very demonstrative, as we would have enough freedom to fit any relatively simple curve such as seen here in the representational distance.

---

> > ### Comment · Reviewer_opdC · 2025-04-07
> >
> > Regarding the match in figure 14 - is there any simple setting in which there is a quantitative match between theory and simulations?

---

> > > ### Author Response · Authors · 2025-04-08
> > >
> > > We found that for the experiment on randomly generated regular tasks, sometimes the theory and experimental curves were very close, when all the settings in the theory, such as the effective learning rates and merging threshold, were set manually to the right values. This was not a very consistent result, however, not all randomly generated tasks gave a smooth enough curve to be able to fit it well. Perhaps when considering more complex tasks, these curves will become smoother and easier to fit, but we have not explored this any further.

---

### Official Review · Reviewer_kzvD · 2025-03-13

**Overall Recommendation:** 5

**Summary:**

This paper explores how neural nets learn automata through training, focusing on a parity task in RNNs, with a short foray into a modular arithmetic task in transformers at the end. They are able to (in a very satisfying way!) theoretically derive equations governing the merging of states in the RNN under some assumptions, which provides a number of useful intuitions for when and why such merging occurs, which explains their experimental results. This analysis shows that the learning occurs in phases, with an initial phase where many states effectively "memorize" distinct sequences of training data. This is followed by a second phase where states which constrain all future outputs similarly merge, corresponding the network internally instantiating a finite-state machine (though non minimal) that generalizes to infinite-length sequences. I feel compelled to also say that aside from the actual content - which I found interesting and thought provoking - this paper is extremely well-written and was a pleasure to read!

**Claims And Evidence:**

The main claims are that

- State merging is what underlies generalization, which they show by interpreting/analyzing the internal states of the RNN and their transitions upon seeing another token of input as an automata. Through training they observe that particular distinct sequences of inputs increasingly cause internal RNN states that are close to eachother (ie. the states merge), and ultimately the number of visited internal states becomes finite, corresponding to infinite-lengthscale generalization. This is a convincing example of their claim, though it is of course a single relatively simple example. To be more specific about this limitation, the tasks explored in this work were all regular, and it's not obvious to me what would happen in the case of e.g. parenthesis balancing or Dyck-languages or other irregular things. Similarly, what occurs in the more general case of probabalistic automata? Still, it's very illustrative and coupled with the local interaction theory is convincing.

- A local interaction theory explains why and when states merge over training. The setup of the theory explores how the distance between  two hidden states associated with two distinct sequences of input changes over gradient descent. The theory shows a number of deep intutions - that states merge when they are associated with the same constraints over all future outputs, that states associated with short inputs won't merge, and that there's a dependence on initialization - all of which is born out in experiment. They perform analysis on their experimental RNN on these points (fig. 6 and 7 and 9). Though there are some assumptions made to make the theory, they discuss them and say them explicitly, so that all seems kosher.

**Essential References Not Discussed:**

I thought the references were reasonably complete, and I appreciated the connection to the neuroscience literature - but just to name some other refs that came to mind the authors might be interested in (I don't necessarily think these need to be included):

- Cleeremans, Axel, David Servan-Schreiber, and James L. McClelland. "Finite state automata and simple recurrent networks." Neural computation 1.3 (1989): 372-381.
Classic work on a similar topic. McClelland has other similar work that more directly looks at fixed point structures in RNNs and interprets them as states of an automata but I can't find that work right now.

- Shai, Adam, et al. "Transformers represent belief state geometry in their residual stream." Advances in Neural Information Processing Systems 37 (2024): 75012-75034.
In that paper they show that the internal states of transformers represent the states of the probabalistic automata (they call it a mixed state presentation) that describes the prediction algorithm over some stochastic process. Similar to the inutition in the paper being reviewed, these states are also those that merge sequences of inputs that constrain the future in the same way.

**Experimental Designs Or Analyses:**

I thought their main RNN experiments, and the experiments on random automata were designed and carried out well.

**Methods And Evaluation Criteria:**

Yes, I find the methods good, and appreciate that they tested on random automata in the appendix.

Regarding the transformer experiments towards the end of the paper - I am unsure why they didn't study the modular addition task which has been shown to have the type of sudden phase-transitiony/grokking behavior that their parity RNN has. Instead they studied a modular subtraction task in a network which showed a smoother transition to generalization. I wonder if the results would bear out in a more clean way in the setting of more sudden grokking as their main example had in their RNN. In general I believe there is a lot more work that could have been done on the transformer - but that is reasonably beyond the scope of the current work (and I look forward to seeing it at some later date).

**Other Comments Or Suggestions:**

I think the plots in figure 9 would look cleaner if you showed fewer x and y marker labels - e.g. you could show 0.2, 0.4, 0.6, 0.8, 1 instead of every .1 interval. or even 0.25 interval

**Other Strengths And Weaknesses:**

I've answered this in responses to other questions.

**Questions For Authors:**

How do you think the analysis, theory, and intuitions of this work apply to non-regular and probabilistic algorithmic tasks?

Would the results of your transformer analysis look different if you analyzed a setting where obvious and sharp phase-transition grokking phenomenon occurs, like in the original Nanda et al work studying modular arithmetic?

**Relation To Broader Scientific Literature:**

This is a strong contribution to the field of interpretability in neural networks. Arguably, the entire issue of interpretability can be framed as figuring out the fundamental link between the continuous (up to floating point precision) dynamics of neural networks and the algorithmic and symbolic nature of what we often think of as performing computation (a similar idea applies to neuroscience). This paper does a good job of pretty directly attacking this fundamental issue. There is a long history on this topic, goign back at least to

**Theoretical Claims:**

Yes, I found the theoretical work elegant and also was explained clearly.

---

> ### Author Rebuttal · Authors · 2025-03-31
>
> Thank you for the feedback and spending your time reviewing our paper. Please find below responses to your comments and the changes we will make to the paper.
>
> ***Questions For Authors***
>
> - The intuitions and theoretical model surrounding state mergers are independent of the task and not affected by averages, so these we would expect to see these in general. For a probabilistic regular task we expect everything to still work via a very similar argument as in the deterministic case. However, for a non-regular task a different mathematical object is needed to represent the learned internal structure of the network, and how representational mergers can result in the development of one is unclear. E.g. it is not possible for representational mergers alone to explain parenthesis balancing, as after all the possible mergers have occurred the automaton is still not finite and so there is no reason to expect generalization to longer sequences. We do wonder if perhaps other types of merging occur due to similar continuity arguments, perhaps in the patterns of maps within architectures more advanced than the RNN explored here. This is why we found the merging of attention patterns in transformers interesting, as it may be related to the development of some kind of mathematical object more advanced than a DFA, but this would require significant further study.
>
> - Note that Nanda et al. used weight decay, which we avoided to focus on induced bias in gradient descent. When we run the experiment on modular addition data without weight decay the transition is not as sharp as in Nanda et al., although it is still sharper than the subtraction task. We find qualitatively similar results as for subtraction, where the number of hidden states ends up increasing overall, and the number of attention patterns first increases and then decreases. We will add this plot to the paper. When using weight decay we get a sharper transition and find significant merging in both the attention patterns and hidden states, although studying regularization is beyond the scope of this work.

---

> > ### Comment · Reviewer_kzvD · 2025-04-04
> >
> > Thank you for this response. I maintain my score of Strong accept.

---

### Official Review · Reviewer_5QF7 · 2025-03-14

**Overall Recommendation:** 4

**Summary:**

This paper studies the learning dynamics in RNNs trained on a toy-task to understand what conditions influence generalization in RNNs. The authors first study RNNs trained on the streaming parity task, and group together RNN hidden states to construct Discrete Finite Automaton (DFA) proxy-models of the RNN. They study how the number of states in the constructed DFAs evolve during training and find that training starts with few states, then over-fitting each data-point with a tree of states. Finally, states merge until the DFA truly becomes finite, at which point generalization occurs. The authors attempt to explain the decrease, leading to generalization, by suggesting that for continuous models the continuity of the model will lead to nearby states merging because continuity will result in the same solution being found. The authors attempt to explain the increase of states, earlier in training, by suggesting that differing effective learning rates lead to states drifting apart. These explanations are justified by analyzing two simplified mathematical models for sequence learning via gradient descent. By way of experiments and simplified model analysis, the authors connect the generalization phenomenon with parameter initialization strength and dataset size.

## Update After Rebuttal
Thank you to the authors for the many clarifications! I am mostly satisfied by their responses and have thus increased my score to a 4. I have chosen not to give the paper a 5 on account of the disconnect between the studied toy model (continuous functions of sequences) and the system of interest (RNNs), and some minor clarity issues (e.g. greater intuition for equations 3, 7).

**Claims And Evidence:**

The claims seem reasonable, except the reviewer would like to see greater discussion of the assertion that continuity is the key property that enables mergers. The reviewer wonders if it might be some other regularity or smoothness property that is crucial (see Question 2 of “Questions for Authors” section).

**Essential References Not Discussed:**

The reviewer cannot immediately think of any missed references.

**Experimental Designs Or Analyses:**

Experimental design and analysis appear sound

**Methods And Evaluation Criteria:**

Evaluation seems sound, but the reviewer did not review the full supplementary section.

**Other Comments Or Suggestions:**

- It would be nice if the authors could discuss their choice of gradient flow as learning dynamics versus something closer to stochastic gradient descent–perhaps a discussion of pros and cons of this choice.
- Many of the figures, in the main body and the supplementary section, are missing a Y-axis label. Please add!
- Line 138 LHS: “them same” => “them the same”
- Line 119 RHS: “As an example for illustration, suppose that two sequences in the dataset agree on target outputs, and one already has the correct predicted output.” The reviewer found this a little confusing. How is it that they can agree on targets but not predicted output? Is the target not what is being predicted? Perhaps this is a problem with the reviewer’s comprehension but some clarity here could be useful.
- It would be really helpful to have some intuitive description, if possible, of the dynamics in Equation 3.
- Some insight into the derivation of Equation 7 would be great.
- Line 305 LHS: “that the first pairs to merge are the ones for which $n - m$ is minimal.” Isn’t Equation 8 a statement about the system at a fixed point? If this is true, why would it explain which pairs merge earlier or later?
- Line 350 RHS: could be good to give a quick definition of what “Regular Random” means.
- Line 436: suggest: “mathematical structures” => mathematical objects
- Line 671: a definition of “accepting state” could be good
- Line 693: reference for Hopcroft’s algo is missing
- Equation 14 lines 8-9: it seems the $2$ in the denominator should be a $4$.
- Notation in section B.1: it could be nice to use something to distinguish the target $y$ values from the predicted ones.
- Line 814: not sure there is supposed to be a 2 in the denominator for $\frac{1}{\tau_h}$
- Could be nice to elaborate on how Equation 19 was derived
- Equation 25 and 26: might suggest different notation for determinant and trace, given similarity with previously used symbols.

**Other Strengths And Weaknesses:**

## Strengths
The paper is nicely written and the approach of coarse-graining with DFAs is very cool and provides great insight into RNN learning dynamics on the streaming parity task. The authors did a great job with study design, to have selected a mathematical model that is so highly simplified but still appears to yield relevant insight. Lastly, the plots are visually appealing and insightful.

## Weaknesses
The reviewer is primarily worried about some of the mathematical assumptions made in the paper (first two bullets below), and also about a potential mistake in Equation 17 (last bullet below):
- The reviewer is concerned about the closeness of hidden states assumptions (see Q1 of “Questions for Authors” section).
- In equation (16) the authors make the assumption that the difference in hidden state representations follows simple linear dynamics (specifically, that the time derivative is proportional to the state value). Given the coupling with the $D_{y_i}$ variable derived in equation (14), this appears to be a not-insignificant approximation. As such, it would be nice to see some justification for this, along with a mention of this in the main body of the manuscript.
- The reviewer was unable to get from line 6 to line 7 in equation (17). It seems that the above linear assumption is being used here but, unless the reviewer is missing something, there appears to be an error where the second term in equation 7 is different from what it should be by a constant factor.

On the whole the reviewer likes the paper, but cannot recommend it for acceptance until their concerns are addressed. If these and the below comments + questions are satisfactorily addressed the reviewer would likely increase their score.

**Questions For Authors:**

1. Do the authors have insight about how the learning dynamics in a more detailed model would be different from their parameterizing the output maps directly with the parameters of the Taylor expansion?
2. Much of the theory rests on the assumption that the RNN states observed in practice are close enough together for the Taylor expansion of Equation 2 to hold. In Fig.10 (Right), and similarly in Figs 15, 16, it seems that–before merging–the merging states actually get rather far apart from each other (looking at the blue trace). Does this not invalidate the closeness assumption? Relatedly, do all states usually get far apart like this before merging, or do some stay close together during the training dynamics?
3. The studied toy dynamical system model seems to provide an argument for the sufficiency of continuity, but perhaps not for the necessity of continuity for this phenomenon. Can the authors think of ways that they could test a discontinuous map (or at least an approximation of such), to determine the necessity of continuity? The reviewer wonders if it might be some form of regularity, rather than the continuity per se, that is the important quality here.
4. In supplementary figure 16 it doesn’t seem like the initial weight scale has much of an effect on “Number of States” or “Merging” with a Tanh, in comparison to the ReLUs used in the paper. How do the authors explain this?
5. Part of the authors’ motivation for the work is from a neuroscience perspective. However, the spike coupling between neurons is often viewed as discontinuous. Do the authors believe that the intuition provided in this paper would not apply to spiking networks?

**Relation To Broader Scientific Literature:**

The reviewer is less familiar with the Discrete Finite Automata (DFA) literature so is ill positioned to discuss this. Conversely, the reviewer is familiar with literature on analysing learning in RNNs. To the reviewer's understanding, much of the recent work on how RNNs learn has focused on computational analysis of internal dynamics (Ostrow et al., 2024, NeurIPS), analysis of linear RNNs (Zucchett & Orvieto, NeurIPS, 2024; Li et al. JMLR, 2022), or analysis of very simple, non-optimal, learning rules (Clark & Abbott, Phys Rev X, 2024). From the reviewer’s point of view, this work is novel because it uses DFA coarse-graining to study the RNN representations algorithmically (although, again, the reviewer is less familiar with DFA literature so such analysis may have appeared there), and because the analytical results deal with a hyper-simplified model–casting aside RNN architecture altogether–and seem to provide some insights that still apply to learning in RNNs.

**Theoretical Claims:**

I am concerned about the assumption made in Equation 16 and a potential error in Equation 17 (see bullet points 2 and 3 in “Other Strengths and Weaknesses” section).

---

> ### Author Rebuttal · Authors · 2025-03-31
>
> Thank you for the feedback and spending your time reviewing our paper. Please find below responses to your comments and the changes we will make to the paper.
>
> ***Weaknesses***
> - We are unsure if this is related to the question about equation (16), but for clarity we would like to note that in equation (16) we mean to assume that $\frac{d}{dt}dh$ and $dh$ are proportional during training only as vectors. They still differ by a time-dependent proportionality coefficient. Complex behavior can happen within this coefficient and $dh$ does not have linear dynamics. We can see this as some of the results, such as a transition between a rich and lazy regime in equation (4) are not present for linear dynamics. We will change the text to make this point clearer.\
> We will also add a mention of this assumption in the main body. In our view, however, it is does not change the bigger picture. We are only restricting to a set of solutions that we can easily solve, which are still valid solutions to the system in equation (14). Solutions that move towards or away from each other, are, due to the importance of mergers in this paper, precisely the ones we would like to study. Additional solutions may exist, where representations spin around each other, but this is less relevant to the formation of finite automata.
> - There are a few steps between line 6 and 7 and the appendix will be changed to explain these in more detail. We use the Ansatz $\frac{\mathrm{d}}{\mathrm{d}t} dh \propto dh \implies \frac{\frac{d}{dt}dh}{||\frac{d}{dt}dh||}=\frac{dh}{||dh||}\implies \frac{d}{dt}dh=\frac{||\frac{d}{dt}dh||}{||dh||}dh$. This allows us to rewrite the second term in line 6 from equation (17): $D_{y_i}\frac{d}{dt}dh=\frac{||\frac{d}{dt}dh||}{||dh||}D_{y_i}dh=\frac{\frac{||\frac{d}{dt}dh||}{||dh||}||dh||^2}{||dh||^2}D_{y_i}dh=\frac{dh^\top (\frac{||\frac{d}{dt}dh||}{||dh||}dh)}{||dh||^2}D_{y_i}dh=\frac{dh^\top\frac{d}{dt}dh}{||dh||^2}D_{y_i}dh$, where we used the previous relation twice.
>
> ***Questions For Authors***
> 1. One can write exact equations of the pair's dynamics by replacing the effective learning rates with time-varying tangent kernels. These depend on the parameterization and thus the architecture. The rest of the equation's structure still is the same. By replacing the kernels with constants, the architecture-independent part of the dynamics could be examined on its own. In a more detailed model we would still see this part of the equation and therefore expect similar merging results, but the kernels may add highly complex behavior.
> 2. The representational distance in the plots 10, 15, 16 are normalized in order to compare the patterns, as distances for the merging pair are much smaller than the diverging pair. The merging pair was reduced by about a factor 100. This is not clear from the figure's description and we will change this.\
> We believe the observation that additional freedom in the model can result in initial divergence patterns is interesting on its own, as it offers an explanation to the tree fitting phase. However, the paper could do better discussing the accuracy of the approximation in the experiment. Even though the representational distance is small this does not guarantee the approximation holds. We ran the experiment and tracked the values of the first and second order term in the Taylor expansion for 100 randomly selected pairs of hidden states. We found that on average for pairs that merge the first order term always dominates by at least a factor of 10. We will add this to the paper.\
> From what we have seen all merging pairs move apart before merging with a similar pattern.
> 3. For the state merger intuitions to hold, nearby hidden states moving closer must result in their respective predictions also moving closer. Continuity provides this, but a weaker condition may be enough. If, for instance, the model is discontinuous but still on average predictions move closer when their hidden states do, the intuitions still make sense. In a neuroscience context this may be an interesting point to mention, so we will add a discussion in the paper. When adding a discontinuous jump in the output map we find a similar automaton development pattern, albeit with noisier dynamics.
> 4. The scale $G$ increases with the initial weight scale, but the rate at which it does will depend on the architectural details. Since a hyperbolic tangent non-linearity saturates and a ReLU does not, how $G$ and thus the occurrence of mergers depends as a function on the initial weight scale may be qualitatively different. In particular for a tanh model near the saturating regime one may expect less dependence on $G$ compared to $N$ which is not affected by architectural choices.
> 5. As mentioned in question 3, discontinuity is still okay as long as on average there is a local relationship between the representations and their predicted outputs. In the case of spiking networks the intuitions may still apply to the firing rates.

---

> > ### Comment · Reviewer_5QF7 · 2025-04-04
> >
> > Thank you to the authors for the detailed response! The reviewer is satisfied with the authors' rebuttal of the *Weaknesses* and *Questions for Authors* section and has raised their score to a 3 accordingly. If the authors sufficiently address the *Other Comments or Suggestions* section in the original review the reviewer will further raise their score to a 4.

---

> > > ### Author Response · Authors · 2025-04-08
> > >
> > > We were unable to go into detail on the "Other Comments Or Suggestions" section before due to character limit, but the suggestions are very much appreciated and we intend to make changes to improve clarity of the paper.
> > >
> > > - We chose gradient flow in our modeling to avoid introducing any form of noise. This is to demonstrate that noise is not a requirement for this form of generalization. We will mention this more explicitly in the text.
> > > - We will add a y-label to every plot that is currently missing one.
> > > - We will change "them same" in line 138 to "them the same".
> > > - The example in line 119 is meant to occur at some point halfway during training, where perhaps some datapoints are already fitted correctly, but others are not. This is not entirely realistic, as in practice predictions are learned simultaneously, but it can still be illustrative to consider as it may help us understand the effect of one representation on another one nearby. We will change the text to make the setting we are considering clearer.
> > > - From the first line of equation (3) we can see that $\langle w_i \rangle_i$ controls the velocity of $||dh||^2$. From the second line we see that the velocity of $\langle ||dy_i||^2 \rangle_i$ is also proportional to $\langle w_i \rangle_i$, although modulated by a positive factor. This means that $||dh||^2$ and $\langle ||dy_i||^2 \rangle_i$ will move in the same direction until $\langle w_i \rangle_i$  decays to zero, at which point they will converge at the same time. A lot of the terms in the third line do not seem to have a clear interpretation, so it is hard to give a very complete intuitive description. In numerical solutions what we see is $\langle w_i \rangle_i$ decaying to zero, but depending on the initialization sometimes it first overshoots zero. This results in $||dh||$ and $\langle ||dy_i||^2 \rangle_i$ changing direction at some point before convergence. When the target outputs agree we see exponential decay.
> > > - There isn't much of a derivation of equation (7), but we can make it clearer by explaining in the text more what is meant by $G$. When applying the recurrent map at initialization it will decrease representational distances by some factor $G$. Thus distances between sequences of length $n$ at initialization  will on average scale as $G^m$ with $G$, where $m$ is the length of the sequence corresponding to the hidden state, and output prediction distances with scale as $G^{n+m}$ where $n$ is the length of the subsequent sequence.
> > > - In line 305 by first we mean the first pairs when one decreases the initialization, not first during training. This is unclear and we will change "first pairs to merge" to "first pairs to merge when decreasing the weight initialization".
> > > - We will add an explanation of how random regular tasks were generated to the paper.
> > > - We will change "mathematical structures" in line 436 to "mathematical objects".
> > > - We will elaborate more on the definition of a DFA and accepting states.
> > > - We will add a reference for Hopcroft's algorithm.
> > > - The $2$ in equation (14) should indeed be a $4$ and we will change this. It does not affect any of the results as this factor of 2 can be absorbed into the arbitrary constant $\frac{1}{\tau_{y_i}}$.
> > > - We will change the target value notation from $y_{\alpha,i}$
> > >  to $y_{\alpha,i}^*$, to distinguish it from predictions.
> > > - As far as we could find there is no calculation error in equation (17) so the factor 2 in line 814 should be there. We will adjust the definition of $\frac{1}{\tau_h}$ to remove it anyway because it will make the equations look nicer, due the other added factor 2 from equation (14).
> > > - The relationship in equation (19) was a guess by looking at the form of equation (18) and not derived from anything.
> > > - We will change the determinant and trace notations to $\text{Tr}(J)$ and $\text{det}(J)$.

---

### Decision · Program_Chairs · 2025-05-01

**Decision:**

Accept (oral)

**Comment:**

All reviewers appreciated this paper’s clear exposition, rigorous analysis, and compelling experiments on how recurrent neural networks solve the streaming parity task by effectively learning and consolidating a finite automaton. Multiple reviewers highlighted the insightful theoretical framework illustrating how pairs of hidden states diverge or converge (merge) under gradient flow, and how this merging drives out-of-distribution generalization. The experiments were found to be well-designed and strongly supportive of the authors’ claims.

Reviewers noted areas that could use clarification. The authors’ rebuttal and plan to revise the paper satisfied most concerns. Overall, the reviewers concur that the work represents a strong contribution to our understanding of how neural networks can learn algorithmic behaviors with finite-state-like structures.